# Version 4 CALIPSO IIR ice and liquid water cloud microphysical properties, Part II: results over oceans

Anne Garnier[1], Jacques Pelon[2], Nicolas Pascal[3], Mark A. Vaughan[4], Philippe Dubuisson[5], Ping Yang[6], and David L. Mitchell[7]

[1]Science Systems and Applications, Inc., Hampton, VA 23666, USA
[2]Laboratoire Atmosphères, Milieux, Observations Spatiales, Sorbonne University, Paris, 75252, France
[3]AERIS/ICARE Data and Services Center, Villeneuve d'Ascq, 59650, France
[4]NASA Langley Research Center, Hampton, VA 23681, USA
[5]Laboratoire d'Optique Atmosphérique, Université de Lille, Villeneuve d'Ascq, 59655, France
[6]Department of Atmospheric Sciences, Texas A&M University, College Station, TX 77843, USA
[7]Desert Research Institute, Reno, NV 89512-1095, USA

*Correspondence to*: Anne Garnier (anne.emilie.garnier@nasa.gov)

## Abstract

Following the release of the Version 4 Cloud-Aerosol Lidar with Orthogonal Polarization (CALIOP) data products from Cloud-Aerosol Lidar and Infrared Pathfinder Satellite Observations (CALIPSO) mission, a new version 4 (V4) of the CALIPSO Imaging Infrared Radiometer (IIR) Level 2 data products has been developed. The IIR Level 2 data products include cloud effective emissivities and cloud microphysical properties such as effective diameter ($D_e$) and water path estimates for ice and liquid clouds. This paper (Part II) shows retrievals over ocean and describes the improvements made with respect to version (V3) as a result of the significant changes implemented in the V4 algorithms, which are presented in a companion paper (Part I). The analysis of the three-channel IIR observations (08.65 µm, 10.6 µm, and 12.05 µm) is informed by the scene classification provided in the V4 CALIOP 5-km cloud layer and aerosol layer products. Thanks to the reduction of inter-channel effective emissivity biases in semi-transparent (ST) clouds when the oceanic background radiance is derived from model computations, the number of unbiased emissivity retrievals is increased by a factor 3 in V4. In V3, these biases caused inconsistencies between the effective diameters retrieved from the 12/10 and 12/08 pairs of channels at emissivities smaller than 0.5. In V4, microphysical retrievals in ST ice clouds are possible in more than 80 % of the pixels down to effective emissivities of 0.05 (or visible optical depth ~ 0.1). For the month of January 2008 chosen to illustrate the results, median ice $D_e$ and ice water path (IWP) are, respectively, 38 µm and 3 g·m$^{-2}$ in ST clouds, with random uncertainty estimates of 50 %. The relationship between the V4 IIR 12/10 and 12/08 microphysical indices is in better agreement with the "severely roughened single column" ice habit model than with the "severely roughened 8-element aggregate" model for 80 % of the pixels in the coldest clouds (< 210 K) and 60 % in the warmest clouds (> 230 K). Retrievals in opaque ice clouds are improved in V4, especially at night and for 12/10 pair of channels, owing to corrections of the V3 radiative temperature estimates derived from CALIOP geometric altitudes. Median ice $D_e$ and IWP are 58 µm and 97 g·m$^{-2}$ at night in opaque clouds, with again random uncertainty estimates of 50 %. Comparisons of ice retrievals with Aqua/Moderate Resolution Imaging Spectroradiometer (MODIS) in the tropics show a better agreement of IIR $D_e$ with MODIS visible/3.7 µm than with MODIS visible/2.1 µm in the coldest ST clouds and the opposite for opaque clouds. In prevailingly supercooled liquid water clouds with centroid altitudes above 4 km, retrieved median $D_e$ and liquid water path are 13 µm and 3.4 g.m$^{-2}$ in ST clouds, with estimated random uncertainties of 45 % and 35 % respectively. In opaque liquid clouds, these values are 18 µm and 31 g.m$^{-2}$ at night, with estimated uncertainties of 50 %. IIR $D_e$ in opaque liquid clouds is smaller than MODIS visible/2.1 and visible/3.7 by 8 µm and 3 µm, respectively.

## 1 Introduction

The Imaging Infrared Radiometer (IIR) is one of the three instruments on board the Cloud-Aerosol Lidar and Infrared Pathfinder Satellite Observations (CALIPSO) satellite which has been in quasi-continuous operation since mid-June 2006 (Winker et al., 2010). IIR is co-aligned with Cloud-Aerosol Lidar with Orthogonal Polarization (CALIOP) and with the Wide Field-of-view Camera (WFC), which are all arranged in a staring, near-nadir looking configuration. The IIR instrument includes three medium resolution channels in the atmospheric window centered at 08.65 µm, 10.6 µm, and 12.05 µm with 1-km$^2$ pixel size. Geo-located and calibrated radiances for all channels are reported in IIR Level 1 products. The IIR Level 2 data products include clouds effective emissivities and cloud microphysical properties such as effective diameters and ice or liquid water path estimates. Following the release of the version 2 IIR Level 1 products (Garmier et al., 2018) and of the version 4 (V4) CALIOP data products, a new version 4 (V4) of the IIR Level 2 data products has been developed and is now available publicly.

The V4 algorithm and its changes with respect to version 3 (V3) are presented in a companion paper (Garnier et al., 2020, hereafter "Part I"). Cloud microphysical properties are derived using the split-window method relying on the analysis of inter-channel effective absorption optical depth ratios, or microphysical indices, from which effective diameter is inferred. The concept of the microphysical index was introduced by Parol et al. (1991) and has been widely used for operational retrievals (Heidinger and Pavolonis, 2009; Pavolonis, 2010). Ice cloud absorption is stronger at 12.05 µm than at 10.6 µm or 08.65 µm. As a result, the brightness temperatures are smaller at 12.05 µm, hence a well know split-window retrieval approach is used in the analysis of inter-channel brightness temperature differences (Inoue, 1985). Hyperspectral infrared sensors such as Atmospheric Infrared Sounder (AIRS) or Infrared Atmospheric Sounder Interferometer (IASI) allow advanced multiple-channel analyses using optimization techniques (Kahn et al., 2014) and the analysis of the spectral coherence of the retrieved cloud emissivities (Stubenrauch et al., 2017). The split-window technique in the thermal infrared spectral domain is very sensitive to the presence of small particles having a maximum dimension smaller than approximately 50 µm in the size distribution (Mitchell et al., 2010). It was shown using the Moderate Resolution Imaging Spectroradiometer (MODIS) thermal infrared bands that observations in this spectral domain are perfectly suited to unambiguously identify the presence of small ice crystals in cold cirrus clouds (Cooper and Garrett, 2010). As such, thermal infrared techniques can provide insights into the observations of small crystals by some in situ instruments when measurements of sizes smaller than 15 µm are uncertain (Mitchell et al., 2018) and help evaluate the possible effects of crystal shattering (Cooper and Garrett, 2011).

Regardless of the retrieval approach, the split-window technique is best adapted for retrievals in clouds of medium effective emissivity. Uncertainties are minimum for cloud effective emissivities between 0.2 and 0.9 (Garnier et al., 2013, hereafter G13), or cloud optical depth between about 0.45 and 4.6, where the information content is the largest (Iwabuchi et al., 2014; Wang et al., 2016). Given sufficiently accurate emissivity estimates, retrievals of cloud properties beyond these lower and upper limits remain possible until the emissivities are either too close to 0 for sub-visible clouds or too close 1 for clouds behaving as blackbody sources, at which points the technique totally loses sensitivity. In addition, the logarithmic relationship between cloud optical depth and infrared emissivity causes a saturation of the cloud optical depths retrievals. For instance, emissivities larger than 0.99 correspond to cloud optical depth larger than only 9. Techniques relying on the combination of visible and near-infrared bands, as used in MODIS operational retrievals (Nakajima and King, 1990; Platnick et al., 2017), are better suited than thermal infrared techniques for cloud optical depths larger than 5 (Wang et al., 2011), but these methods are limited to daytime observations only.

Due to its sensitivity to small particles, the split-window technique is an attractive option for retrievals of liquid droplets sizes (Rathke and Fisher, 2000), and microphysical retrievals in liquid water clouds are now included in the V4 IIR products. All other things being equal, the performance of the split-window technique increases with the radiative contrast between the cloud and the surface. Consequently, retrieval uncertainties are larger for liquid water clouds, which typically form relatively close to the Earth's surface, and hence these retrievals were not included in V3. Liquid water clouds such as marine stratocumulus clouds, which are an important component of the Earth system, have optical depths typically larger than 10, well beyond the range of applicability of the technique. However, infrared observations have the potential to provide new insight into the microphysical properties of thin liquid water clouds (Turner et al., 2007; Marke et al., 2016) and of supercooled mid-level liquid water clouds.

The IIR analyses start with the retrieval of cloud effective emissivities in each channel, which are then converted to effective absorption optical depths as $\tau_{a,k} = -\ln(1 - \varepsilon_{eff,k})$, where $\varepsilon_{eff,08}$, $\varepsilon_{eff,10}$, and $\varepsilon_{eff,12}$ are the effective emissivities retrieved in IIR channels 08.65 (k = 08), 10.6 (k = 10), and 12.05 (k = 12), respectively. Effective emissivity is mostly a measure of cloud absorption, and the term "effective" refers to the contribution from scattering, which is the most significant at 08.65 µm. The first IIR microphysical index, $\beta_{eff}12/10 = \tau_{a,12}/ \tau_{a,10}$, is the ratio of the effective absorption optical depths at 12.05 and 10.6 µm and the second one, $\beta_{eff}12/08$ = $\tau_{a,12}/ \tau_{a,08}$, is the ratio of the effective absorption optical depths at 12.05 and 08.65 µm. Two main pieces of information are needed to retrieve these quantities: the cloud Top Of Atmosphere (TOA) blackbody radiance, which requires a good estimate of the cloud radiative temperature, and the TOA background radiance that would be observed if no cloud were present. The former drives the accuracy at large emissivities and the latter the accuracy at small emissivities.

The first step into any retrieval approach is the detection of a cloud and the determination of its thermodynamic phase and radiative temperature. The ability to ascertain cloud amounts and characteristics varies with the observing capabilities of different passive sensors (Stubenrauch et al., 2013). Even though IIR has only three medium resolution channels, its crucial advantage is the quasi-perfect co-location with CALIOP observations. Indeed, as emphasized by Cooper et al. (2003), cloud boundaries measured by active instruments provide an invaluable piece of information for obtaining accurate estimates of cloud radiative temperatures. The IIR algorithm relies on CALIOP's highly sensitive layer detection to characterize the atmospheric column seen by each IIR pixel. CALIOP provides geometrical altitudes, which are converted into radiative temperatures. The radiative temperature, $T_r$, of a multi-layer cloud system is estimated as the thermodynamic temperature, $T_c$, at the centroid altitude of the CALIOP attenuated backscatter at 532 nm. In the V4 algorithm, this estimate is further corrected when single or multi-layer ice cloud systems are observed (Part I). The thermodynamic temperature is derived from interpolated temperatures profiles of the Global Modeling and Assimilation Office (GMAO) Modern-Era Retrospective analysis for Research and Applications, Version 2 (MERRA-2) model (Gelaro et al., 2017).

The second retrieval step is the determination of the TOA background radiance, which often requires simulations using ancillary meteorological profiles and surface data. These simulations are generally more accurate over oceans than over land because the surface emissivities in the various channels are better known and less variable over oceans, and the skin temperature data are usually more accurate. In this paper, we therefore focus on retrievals over oceans. In the IIR algorithm, the TOA background radiance is preferentially determined using observations in neighboring pixels in those cases when clear sky conditions, as determined by CALIOP, can be found. Otherwise, it is computed using the FASRAD radiative transfer model (Garnier et al., 2012; Dubuisson et al., 2005). In V3, IIR microphysical retrievals over oceans were possible down to $\varepsilon_{eff,12} \sim 0.05$ (or optical depth $\sim$ 0.1) when the background radiance could be measured in neighboring pixels (G13). When the background radiance had to be computed by FASRAD, which represents about 75 % of the cases, inter-channel biases in the model simulations caused discernable flaws in the microphysical retrievals. The inter-channel biases in the FASRAD simulations have been significantly reduced in V4, as discussed in Part I.

This paper aims at demonstrating the improved accuracy of the V4 effective emissivities and of the subsequent microphysical indices that result from the changes implemented in the V4 algorithm (Part I), and at illustrating the changes in the retrieved microphysical properties. Our assessment is carried out after carefully selecting the relevant cloudy scenes, following the rationale presented in Sect. 2. Retrievals in ice clouds are presented in Sect. 3, which includes step by step comparisons between V3 and V4, examples of V4 retrievals, and comparisons with MODIS retrievals. Section 4 is dedicated to retrievals in liquid water clouds that were added in V4, and Sect. 5 concludes the presentation.

## 2 Cloudy scenes selection

The analysis of the IIR observations is informed by the scene classification provided by the V4 CALIOP cloud and aerosol 5-km layer products. This scene classification is established for layers detected by the CALIOP algorithm at 5-km and 20-km horizontal averaging intervals (Vaughan et al., 2009). An example is shown in Fig. 1, which was extracted from nighttime granule 2008-01-30T09-15-45ZN on January 30th, 2008.

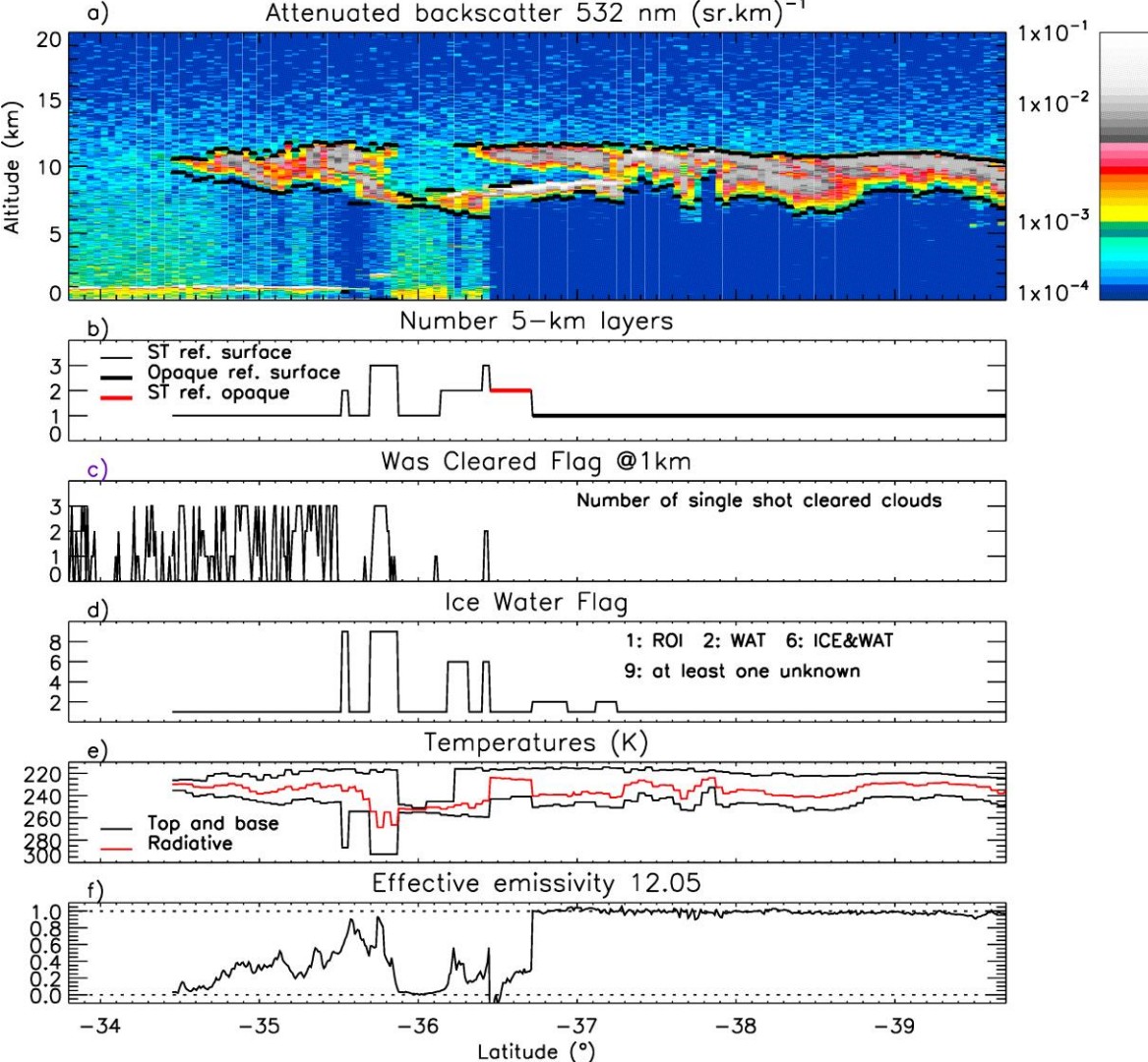

**Figure 1: Example of the CALIOP scene classification information used for effective emissivity retrievals on January 30th, 2008 (granule 2008-01-30T09-15-45ZN). (a) CALIOP attenuated backscatter with top and base altitudes of the cloud-system highlighted in black; (b): number of cloud layers in the cloud-system; cases with Earth surface as a reference are denoted with black lines (thin: semi-transparent (ST) layers; thick: 1 opaque layer) and in red are the cases with the lowest opaque cloud as a reference; (c): CALIOP "Was Cleared Flag" at 1-km IIR pixel resolution; (d): Ice Water Flag of the cloud system (e): temperatures at cloud top and cloud base (black) and radiative temperature used by the IIR algorithm (red); (f): effective emissivity of the cloud-system at 12.05 μm. See text for details.**

Figure 1a shows the Level 1 CALIOP attenuated backscatter averaged at 5-km horizontal resolution with the top and base altitudes of the cloud-system shown in black. Cloudy scenes can include one or several layers (Fig. 1b). When the lowest of at least two layers is opaque to CALIOP, this opaque layer is used as a reference assuming it behaves as a blackbody source and the algorithm retrieves the properties of the overlying semi-transparent (ST) layers. An example is found between latitudes - 36.45° and - 36.7°, highlighted in red in Fig. 1b, where the algorithm retrieves the properties of two ST layers overlying the opaque cloud located at about 8 km altitude. South of - 36.7° and down to - 37.2°, the portion of this cloud which is used as an opaque reference between - 36.45°and - 36.7° is included in a single opaque cloud of top altitude equal to 11.5 km, which extends down to the southernmost latitudes. North of -36.45° and up to -34.45°, the atmospheric column includes 1 to 3 semi-transparent clouds. Finally, no cloud layers are seen north of -34.45° where the scenes contain only low ST non-depolarizing aerosol layers (not shown). The atmospheric column might also contain clouds having top altitudes less than 4 km that are detected at single-shot resolution and then cleared before searching for the more tenuous layers typically reported in the 5-km products (Vaughan et al., 2009). These single shot detections are not included in Fig. 1b. The number of these single shot cleared clouds seen within each IIR pixel is shown in Fig. 1c. We showed in Part I (Fig. 5 in Part I) that the presence of these cleared clouds modifies the background radiance compared to the radiance due to the ocean surface and ultimately biases the effective emissivity retrievals. Because these biases cannot be quantified a priori, scenes that contain single shot cleared clouds should be treated with caution. The Ice Water Flag shown in Fig. 1d characterizes the ice/water phase of the cloud layers included in the cloud system. These layers are classified either as ice, liquid water, or "unknown" by the V4 CALIOP Ice/Water phase algorithm (Avery et al., 2020). Most of the ice clouds are composed of randomly oriented ice (ROI) crystals. Clouds containing significant fractions of horizontally oriented ice (HOI) crystals are also detected, mainly before the end of November 2007, when the platform tilt angle was changed from its initial 0.3° orientation to a view angle of 3° (Avery et al., 2020). In Fig. 1 we find cloud systems composed of ROI only (flag = 1), liquid water (WAT) only (flag = 2), ice and WAT (flag = 6), and some systems that include at least one layer of unknown phase (flag = 9). IIR effective emissivities are reported for all single or multi-layer scenes, regardless of the phase. In V4, the phase information is used to adjust the radiative temperature (Fig. 1e) estimates in cases containing ice clouds (Part I). For illustration purposes, the V4 retrieved effective emissivities at 12.05 µm are shown in Fig. 1f. In this example, emissivity values in the opaque cloud are mostly around 1, the lowest value being 0.91 at - 39.5° where the CALIOP image suggests the presence of a faint signal below the cloud. Effective emissivities in ST clouds vary between 0 and 0.9. The only exception is between -36.45° and -36.52°, where non-physical negative effective emissivities are retrieved because the computed background radiances are smaller than the observed radiances, and are therefore underestimated. In this case, the reference is a cloud classified as opaque by CALIOP (see area highlighted in red in Fig. 1b), which is likely not sufficiently dense to behave as a blackbody reference.

This example shows that a cloudy scene can include a variety of conditions for the IIR retrievals. Because the goal here is to present the cloud microphysical properties as retrieved with the IIR V4 algorithm and improvements with respect to V3, we chose to limit the analyses to scenes that contain only ROIs, only HOIs, or only WAT clouds with background radiances from the ocean surface. Furthermore, in order to facilitate the interpretation of the results, we require that the CALIOP cloud-aerosol discrimination algorithm (Liu et al., 2019) assign high confidence to the cloud classifications and likewise that the ice/water phase algorithm determined the phase classifications with high confidence. Finally, scenes containing single shot cleared clouds are discarded. Table 1 reports the fraction of scenes that fall into these categories. The statistics are for IIR pixels between 60° S and 60° N in January and July 2008. The ROI scenes represent 13 % to 16 % of all the IIR pixels. The HOI scenes represent less than 0.1 % of all the IIR pixels, and we found that they represent less than 1 % at the beginning of the mission when the platform tilt angle was

0.3°. Thus, in the rest of the paper, ice clouds will refer to scenes containing only ROI layers. The WAT scenes represent 14 to 19 % of all the IIR pixels.

Clear sky conditions are defined as cloud free scenes with Was Cleared Flag at 1 km resolution equal to zero, with no aerosol layers or only low (< 7 km) semi-transparent "not dusty" layers. Dusty layers are those identified as dust, polluted dust, or dusty marine (Kim et al., 2018) and are discarded because they may have a signature in the IIR channels (Chen et al., 2010). For comparison with the previous categories, the clear sky conditions represent 20 % of the cases for daytime data and 15 % for nighttime data. It is noted that 6 to 10 % of the pixels are rejected as "clear sky" in V4 due to the presence of single shot cleared clouds. These pixels would have been accepted by the V3 algorithm: they represent 25 % and 35 % of the V3 clear sky conditions for daytime and nighttime data, respectively.

Table 1: Total number of IIR pixels, fraction of IIR pixels with only high confidence ROI, WAT, and HOI layers in the column and no single shot cleared clouds for retrievals with background radiance from ocean surface between 60° S and 60° N, and fraction of clear sky pixels.

| | January 2008 | | July 2008 | |
|---|---|---|---|---|
| | Night | Day | Night | Day |
| # IIR pixels | $4.2 \times 10^6$ | $4.2 \times 10^6$ | $3.8 \times 10^6$ | $3.9 \times 10^6$ |
| | Fraction of IIR pixels | | | |
| ROI | 0.132 | 0.160 | 0.127 | 0.155 |
| HOI | < 0.001 | < 0.001 | < 0.001 | < 0.001 |
| WAT | 0.175 | 0.192 | 0.143 | 0.182 |
| Clear sky | 0.143 | 0.204 | 0.165 | 0.208 |
| Clear sky rejected in V4 | 0.083 | 0.063 | 0.097 | 0.074 |

Scenes composed of only high confidence ROI layers or only WAT layers can include either one opaque layer or a number of ST layers. This is quantified in Table 2 for the months of January and July 2008. For these months, 45 to 53 % of the selected ROIs are opaque to CALIOP while opaque clouds represent 67 to 90 % of the WATs. Daytime fractions of opaque clouds are larger than nighttime ones, which is likely due daytime surface detection issues. Scenes with only ST layers are spread into three main categories: only one layer, two vertically overlapping layers detected at different horizontal averaging resolutions where the top altitude of the lower layer is greater than the base altitude of the higher layer, and multi-layer configurations with two non-overlapping layers or more than two layers. For both ROI and WAT clouds, the vast majority of the ST scenes have only one layer in the column, which is explained by the fact that we required all the layers to be characterized with high confidence. Thus, the study will be carried out for single-layer cases for simplicity.

Table 2: Detailed statistics for IIR pixels with only ROI or WAT high confidence layer(s) in the column and no cleared clouds for retrievals with background radiance from ocean surface between 60° S and 60° N in January and July 2008: fraction of opaque clouds, single-layered ST clouds, ST clouds with two overlapping layers, and multi-layered ST clouds.

| | ROI | | | | WAT | | | |
|---|---|---|---|---|---|---|---|---|
| | January 2008 | | July 2008 | | January 2008 | | July 2008 | |
| | Night | Day | Night | Day | Night | Day | Night | Day |
| Opaque | 0.452 | 0.487 | 0.470 | 0.533 | 0.786 | 0.899 | 0.672 | 0.864 |
| ST one layer | 0.494 | 0.458 | 0.482 | 0.420 | 0.200 | 0.097 | 0.313 | 0.131 |

| ST overlap | 0.007 | 0.006 | 0.008 | 0.004 | 0.006 | <0.001 | 0.007 | < 0.001 |
| ST multi-layers | 0.047 | 0.049 | 0.040 | 0.043 | 0.008 | 0.003 | 0.008 | 0.004 |

## 3 Retrievals in ice clouds

The accuracy of the effective emissivity in each IIR channel and of the subsequent microphysical indices is a prerequisite for successful retrievals of cloud microphysical properties. In section 3.1, we use internal quality criteria to demonstrate the improvements in the V4 effective emissivities in ice clouds that result from the revised computed background radiances over oceans and from the revised radiative temperature estimates (Part I). After examining the changes in $\varepsilon_{eff,12}$ (at 12.05 µm), inter-channel effective emissivity differences, $\Delta\varepsilon_{eff}12\text{-}k = \varepsilon_{eff,12} - \varepsilon_{eff,k}$, are assessed, keeping in mind that they should tend towards zero on average when $\varepsilon_{eff,12}$ tends towards 0 and towards 1 (G13; Part I). Changes in the visible cloud optical depth, $\tau_{vis}$, inferred from the summation of absorption optical depths at 12.05 µm and 10.6 µm ($\tau_{a,12} + \tau_{a,10}$, Part I) are shown in Sect. 3.2.

The subsequent improvements in the microphysical indices and in the performance of the microphysical algorithm are discussed in section 3.3, where we also illustrate changes in the effective diameters ($D_e$) reported in V3 and V4. We recall that $D_e$ is defined as $D_e = (3/2) \times (V/A)$, where V is the total volume of the size distribution and A is the corresponding projected area (Foot, 1988; Mitchell et al., 2002). The V4 algorithm uses two ice habit models from the "TAMUice2016" data base (Bi and Yang, 2017; Yang et al., 2013), namely the severely roughened solid column (SCO) and severely roughened 8-element column aggregate (CO8) models, and the model used for the retrievals is selected according to the relationship between $\beta_{eff}12/10$ and $\beta_{eff}12/08$. IIR retrieved $D_e$ is the mean of the $D_e12/10$ and $D_e12/08$ effective diameters when these two values can be retrieved from the respective $\beta_{eff}12/k$; that is, $D_e = \left(D_e12/10 + D_e12/08\right)/2$. Both $D_e12/10$ and $D_e12/08$ are reported in the product for users interested in specific analyses. The V4 look-up tables (LUTs) that relate microphysical index and effective diameter are computed using the FASDOM (Dubuisson et al., 2008) model and bulk single scattering properties derived using an idealized gamma particle size distribution. In V3, the LUTs were derived using single scattering properties of the "solid column" and "aggregate" ice habit models from the database described in Yang et al. (2005), with no particle size distribution. We showed in Part I that everything else being equal, the size distribution introduced in V4 increases retrieved $D_e$. As illustrated in Part I, the microphysical indices are very sensitive to $D_e$ smaller than 50 µm and the sensitivity decreases progressively up to $D_e = 120$ µm which is considered the sensitivity limit of our retrievals in ice clouds.

In Sect. 3.4, we show examples of V4 $D_e$ and Ice Water Path (IWP) microphysical retrievals and comparisons with MODIS retrievals are presented in Sect. 3.5.

### 3.1 Effective emissivity: V4 vs. V3.

Because of numerous changes in the CALIOP V4 algorithms, the cloud layers reported in the V3 and V4 CALIOP data products are not identical, so that direct comparisons of the V3 and V4 IIR data products could be misleading. In order to isolate the changes due to the IIR algorithm, the V3 emissivities (hereafter V3_comp) for clouds reported in CALIOP V4 were recomputed using the V3 computed background radiances reported in the V3 product and the V3-like blackbody temperatures derived directly from the centroid temperatures, $T_c$, which are available in the V4 product along with the V4 blackbody temperatures. The exercise was carried out for V4 scenes over oceans that contain one single cloud layer classified as high confidence ROI with no cleared cloud, as discussed previously in Sect. 2. Illustrations are shown for the month of January 2008 between 60°S and 60° N.

### 3.1.1 Effective emissivity in channel 12.05

The nighttime (blue) and daytime (red) distributions of $\varepsilon_{eff,12}$ are shown in Fig. 2, where V4 (solid lines) is compared with V3_comp (dashed lines). Figs. 2a and 2b show the distributions for ST and opaque clouds, respectively. The V4 median random uncertainty estimates shown in Fig. 2c and 2d are of the order of 0.015 at $\varepsilon_{eff,12} < 0.6$ and increase up to 0.03 at the largest emissivities, where the uncertainty in $\varepsilon_{eff,12}$ is prevailingly due to the uncertainty in the radiative temperature taken equal to $\pm$ 2 K. (Part I). Because of retrieval errors, $\varepsilon_{eff,12}$ can be found outside the range of physically possible values (i.e., 0 to 1). For ST clouds (Fig. 2a), the V3 and

the V4 histograms differ mostly at $\varepsilon_{eff,12} < 0.05$, where the changes in the background radiances have the largest impact. In this example, the fraction of ST clouds with negative $\varepsilon_{eff,12}$ values is reduced from 12 % in V3 to 3.5 % in V4. For opaque clouds (Fig. 2b), the larger V4 $\varepsilon_{eff,12}$ values are due to the radiative temperature corrections introduced in the V4 algorithm (these corrections have essentially no impact for ST clouds). For the range of $\varepsilon_{eff,12}$ values found in opaque clouds, the corrections are prevailingly a function of the "apparent" cloud thickness, which is larger and closer to the true geometric thickness at night (Part I). Nighttime

and daytime $\varepsilon_{eff,12}$ distributions peak at larger $\varepsilon_{eff,12}$ in V4 ($\varepsilon_{eff,12}$ = 0.99 and 0.97, respectively) than in V3 ($\varepsilon_{eff,12}$ = 0.94). Consequently, random uncertainties and possible overcorrections cause an increase of the fraction samples with $\varepsilon_{eff,12} > 1$, from 3 % in V3 to 12 % in V4 at night, and from 1.2 to 3.3 % for daytime data. At night, 98 % of the opaque clouds have V4 $\varepsilon_{eff,12} > 0.8$, or cloud optical depth > 3.2. This lower range of optical depths is consistent with V4 CALIOP optical depth retrievals, even though it is recognized that direct comparisons with V4 CALIOP optical depths in opaque clouds are difficult (Young et al., 2018).

Nighttime $\varepsilon_{eff,12}$ distributions for ST and opaque clouds are essentially mutually exclusive, with a $\varepsilon_{eff,12}$ threshold around 0.7. In contrast, these distributions overlap between 0.4 and 0.7 for daytime data. The tail down to $\varepsilon_{eff,12}$ = 0.4 ($\tau_{vis}$ ~ 1) for daytime opaque clouds data is explained by a greater difficulty for the CALIOP algorithm to detect faint surface echoes during the day due to large solar background noise, so that some clouds of moderate emissivity may be misclassified as opaque by CALIOP. Effective emissivities close to 1 are found in clouds where the CALIOP integrated attenuated backscatter (IAB) is larger than 0.04 sr$^{-1}$, which

is in the upper range of values typically observed in opaque ice clouds (Young et al., 2018). Platt et al. (2011) showed that these large IABs, which are often coupled with small apparent geometric thicknesses, are observed when the CALIPSO overpass is close to the center of a mesoscale convective system. Using cloud retrievals based on AIRS thermal infrared data, Protopapadaki et al. (2017) demonstrated that emissivities close to 1 in the tropics are most often indicative of convection cores reaching the upper troposphere, which confirms our observations based on CALIPSO.

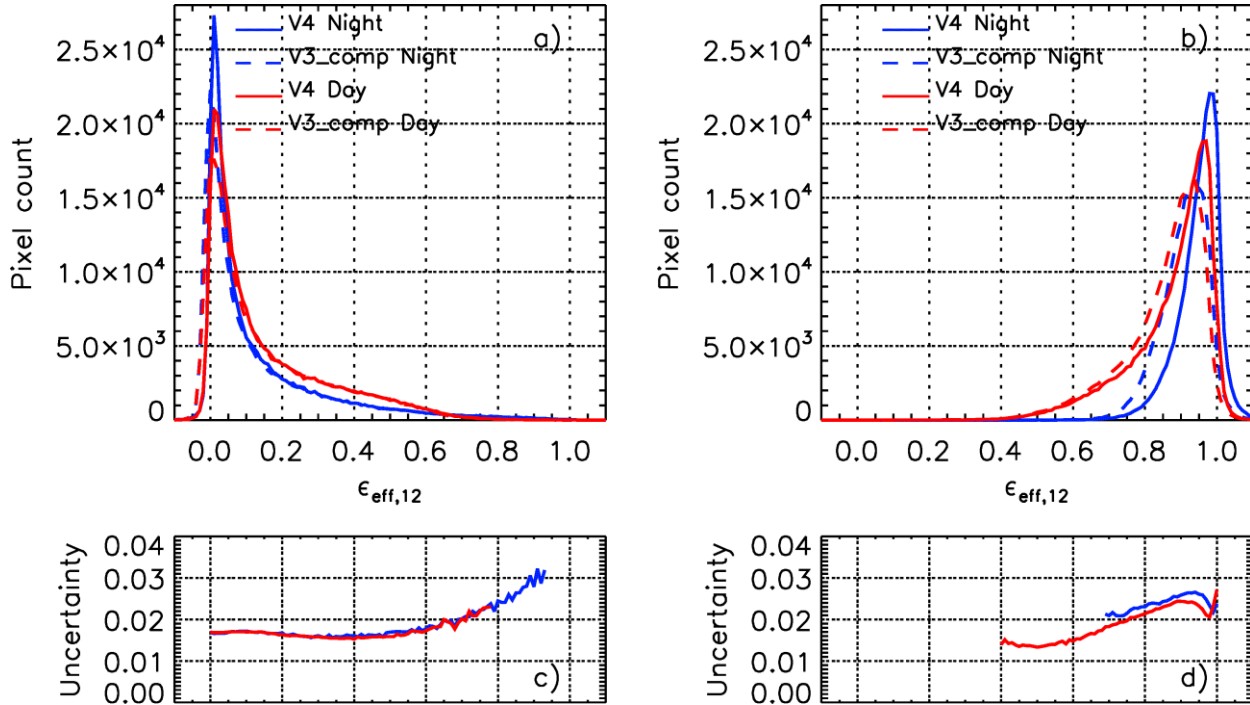

Figure 2: Effective emissivity distributions at 12.05 μm in (a) ST and (b) opaque single-layered ice clouds over oceans between 60° S and 60° N in January 2008 in V4 (solid lines) and in V3_comp (dashed lines). The blue and red curves are for nighttime and daytime data, respectively. Panels (c) and (d) are the V4 median random uncertainty estimates corresponding to panels (a) and (b), respectively.

### 3.1.2 Inter-channel effective emissivity differences

We recall that effective emissivity retrievals preferably use background radiances observed in neighboring clear sky pixels and otherwise use radiances computed by FASRAD. In order to evaluate V4 computed background radiances, we first examined $\Delta\varepsilon_{eff}$12-k at $\varepsilon_{eff,12}$ ~ 0 in ST clouds by separating retrievals that used observed radiances (V4_obs) and those that used computed radiances (V4_comp). The results are reported in Table 3, where $\Delta\varepsilon_{eff}$12-k at $\varepsilon_{eff,12} \approx 0$ is also reported for V3_comp for reference. As in V3 (G13), V4 inter-channel biases are minimum when the background radiance can be determined from observations (V4_obs), which represents 30 % of the retrievals in ST clouds for this dataset. When the background radiance is computed (V4_comp, 70 % of the cases), median $\Delta\varepsilon_{eff}$12-k is similar for both channel pairs and smaller than 0.0025 in absolute value. This indicates residual inter-channel biases smaller than 0.1 K in V4 according to the simulations shown in Fig. 1b of Part I, which is consistent with the residual inter-channel differences seen in clear sky conditions (Part I). Because these biases are very small, retrievals using computed and observed radiances are consistent in V4, hereafter the two methods will be referred to collectively as "V4" for clarity. The $\Delta\varepsilon_{eff}$12-k differences were unambiguously too low in V3_comp, especially for the 08-12 pair, so that reliable retrievals were possible only when observed radiances were available (G13). Including retrievals using computed radiances in V4 increases the number of retrievals in ST clouds by a factor 3.3.

Table 3: Inter-channel effective emissivity differences at $\varepsilon_{eff,12}$ ~ 0 for retrievals in single-layered ST ice clouds over oceans between 60°S and 60°N in January 2008.

| | Fraction of retrievals | | | $\Delta\varepsilon_{eff}$ (12-10) $-0.005 < \varepsilon_{eff,12} < 0.005$ | | $\Delta\varepsilon_{eff}$ (12-08) $-0.005 < \varepsilon_{eff,12} < 0.005$ | |
|---|---|---|---|---|---|---|---|
| | Night | Day | | Night | Day | Night | Day |
| V4_obs | 0.27 | 0.33 | Median | 0.0000 | 0.0002 | 0.0005 | 0.0008 |

| | | | | | | | |
|---|---|---|---|---|---|---|---|
| | | | 25th | -0.003 | -0.003 | -0.002 | -0.002 |
| | | | 75th | 0.003 | 0.003 | 0.003 | 0.004 |
| V4_comp | 0.73 | 0.67 | Median | -0.001 | 0.001 | -0.0025 | 0.0004 |
| | | | 25th | -0.004 | -0.002 | -0.0053 | -0.003 |
| | | | 75th | 0.002 | 0.005 | 0.0003 | 0.0045 |
| V3_comp | N/A | N/A | Median | -0.006 | -0.004 | -0.018 | -0.015 |
| | | | 25th | -0.009 | -0.007 | -0.023 | -0.021 |
| | | | 75th | -0.003 | -0.0001 | -0.014 | -0.010 |

The variations with $\varepsilon_{eff,12}$ of the $\Delta\varepsilon_{eff}12$-k inter-channel effective emissivity differences for the 12-10 and 12-08 pairs are shown in Figs. 3a and 3b, respectively. The curves are median values, and the shaded gray areas are between the V4 nighttime 25th and 75th percentiles. The first observation is that median $\Delta\varepsilon_{eff}12$-k are larger in V4 (solid lines) than in V3_comp (dashed lines) at any emissivity. When $\varepsilon_{eff,12}$ tends towards 1, $\Delta\varepsilon_{eff}12$-k is minimum at $\varepsilon_{eff,12}$ corresponding to the peak of the distributions shown in Fig. 2, which suggests that the peaks should be closer to $\varepsilon_{eff,12} = 1$. This shows that V4 is improved compared to V3, more convincingly for nighttime data, but also that the radiative temperature corrections are likely not sufficient. Consistent with the simulations shown in Fig. 1 of Part I, $\Delta\varepsilon_{eff}12$-k are increased from V3 to V4 at large emissivities, because the radiative temperatures are increased, and the changes are more important in the 12/08 pair than in the 12/10 pair.

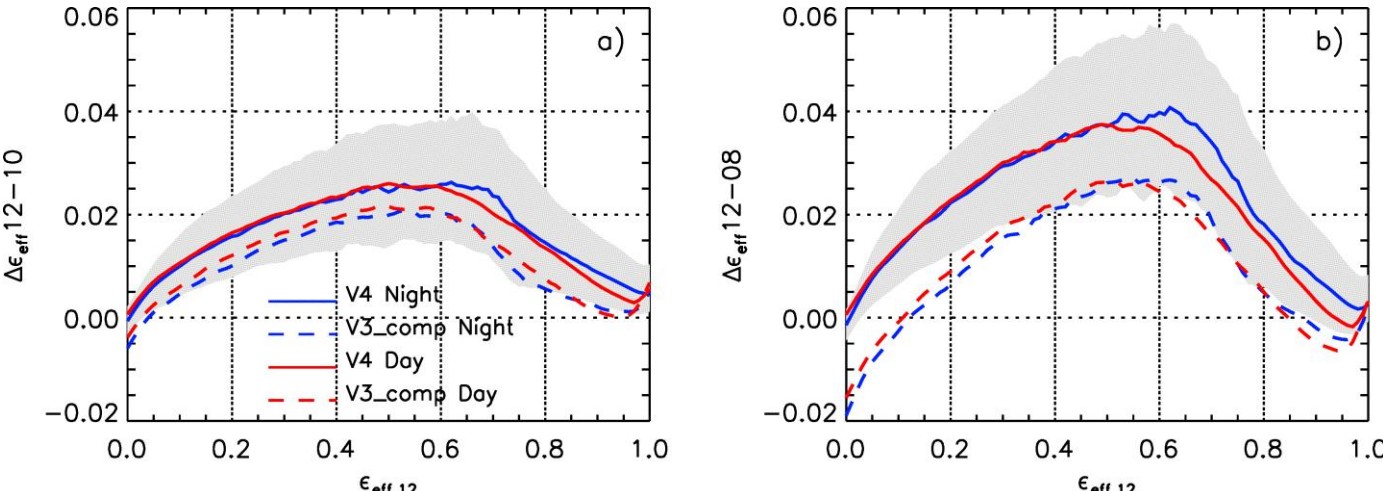

Figure 3: IIR inter-channel (a) $\Delta\varepsilon_{eff}12$-10 and (b) $\Delta\varepsilon_{eff}12$-08 effective emissivity differences vs. effective emissivity at 12.05 µm in single-layered ice clouds over oceans between 60° S and 60° N in January 2008 in V4 (solid lines) and in V3_comp (dashed lines). The blue and red curves are median values for nighttime and daytime data, respectively. The shaded gray areas are between the V4 nighttime 25th and 75th percentiles.

### 3.2 Visible cloud optical depth: V4 vs. V3

The V3-V4 changes in the visible cloud optical depths inferred from $\varepsilon_{eff,12}$ and $\varepsilon_{eff,10}$ are shown in Figs. 4a and 4b for nighttime and daytime data, respectively. The large plots where $\tau_{vis}$ ranges between 0 and 15 are built using bins equal to 0.2, and the embedded small plots show details for $\tau_{vis}$ smaller than 1 and bins equal to 0.02. The changes in $\tau_{vis}$ are smaller than 0.02 on average and not significant for $\tau_{vis}$ smaller than 2 (or $\varepsilon_{eff,12} < \sim 0.6$), that is for most of the ST clouds. For $\tau_{vis} > 2$, V4 $\tau_{vis}$ is increasingly larger than

V3 $\tau_{vis}$, owing to the warmer radiative temperature estimates in V4. Consistent with previous observations regarding $\varepsilon_{eff,12}$, the $\tau_{vis}$ increase from V3 to V4 is larger at night (Fig. 4a) than during the day (Fig. 4b).

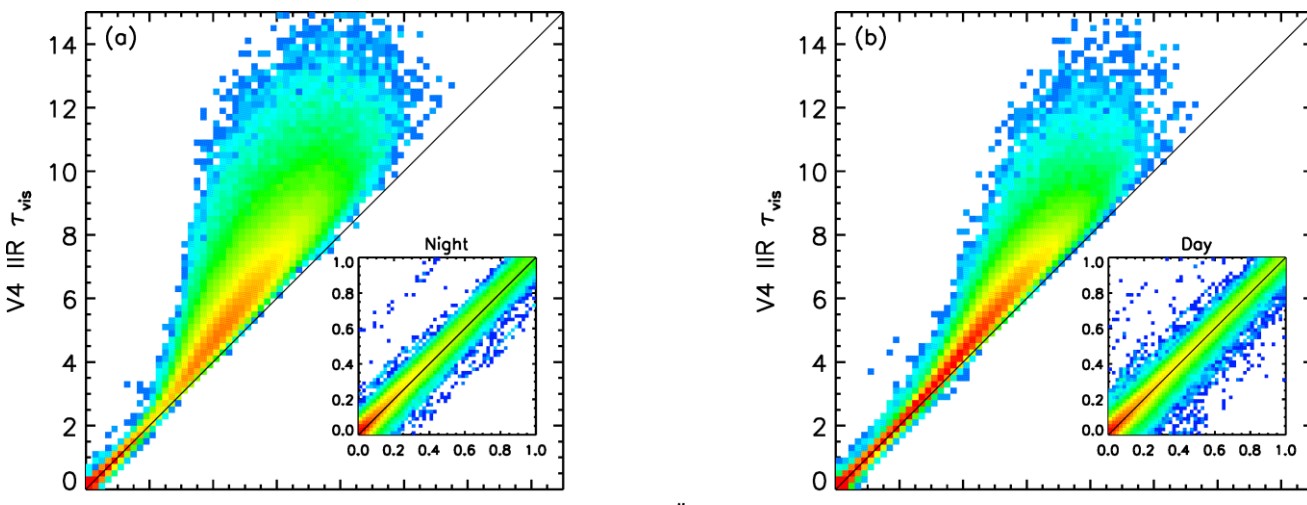

**Figure 4: (a) Nighttime and (b) daytime comparisons of V3 and V4 IIR cloud optical depth ($\tau_{vis}$) in single-layered ice clouds over oceans between 60° S and 60° N in January 2008. The embedded small plots show details for $\tau_{vis}$ between 0 and 1.**

### 3.3 Microphysical indices and effective diameters retrievals: V4 vs. V3

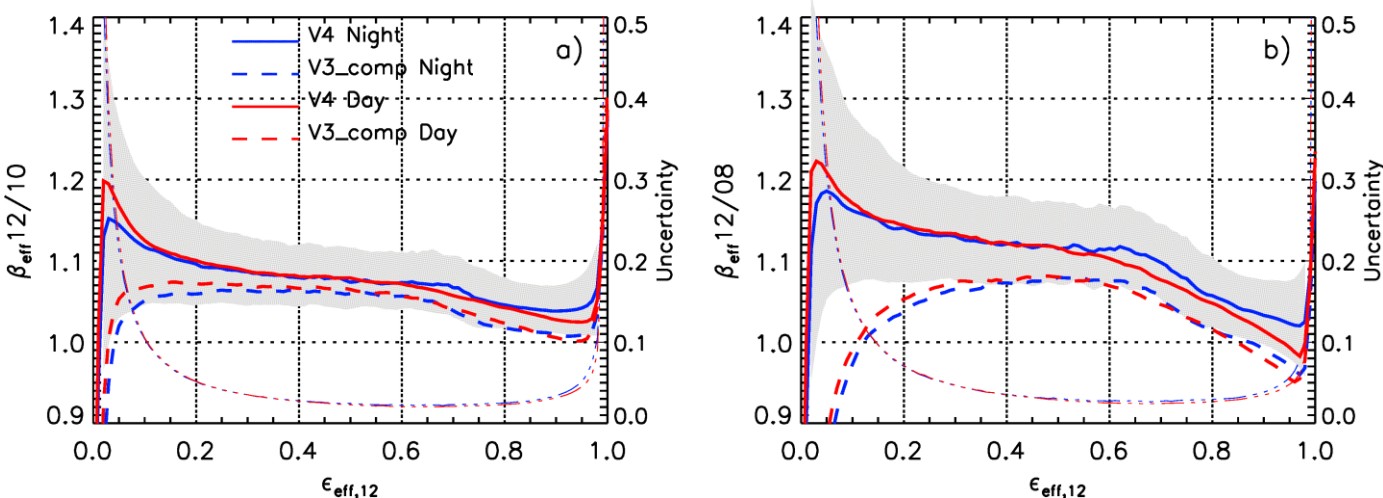

**Figure 5: (a) $\beta_{eff}$ 12/10 and (b) $\beta_{eff}$ 12/08 microphysical indices vs. effective emissivity at 12.05 μm in single-layered ice clouds over oceans between 60°S and 60°N in January 2008 in V4 (solid lines) and in V3_comp (dashed lines). The blue and red curves are the median values for nighttime (blue) and daytime (red) and the shaded gray areas are between the nighttime 25th and 75th percentiles. The blue (night) and red (day) thin dotted dashed lines are the V4 random absolute uncertainty estimates with vertical axis on the right-hand side of each panel.**

The changes in the $\beta_{eff}$12/10 and $\beta_{eff}$12/08 microphysical indices resulting from the changes in $\Delta\varepsilon_{eff}$12-10 and $\Delta\varepsilon_{eff}$12-08 (Fig. 3) are illustrated in Figs. 5a and 5b. The sharp variations of the V4 median microphysical indices (solid lines) at $\varepsilon_{eff,12} < 0.03$ and $\varepsilon_{eff,12} > 0.96$ are due to the increasing truncation of the distributions, because both $\beta_{eff}$12/k indices can be computed only when $0 < \varepsilon_{eff,k} < 1$ in the three channels. Over-plotted in Fig. 5 are the median V4 random absolute uncertainty estimates, which are minimum

and around 0.02 for intermediate emissivity values (G13). The noticeable large dispersion of the $\beta_{eff}12/k$ values at $\varepsilon_{eff,12} < 0.1$ is largely explained by the random uncertainties. The median $\beta_{eff}12/k$ values are overall larger in V4 than in V3_comp, with larger changes for the 12/08 pair than for the 12/10 pair. The consequences for the $D_e$ retrievals are twofold. First, the fraction of $\beta_{eff}12/k$

values that are larger than the low sensitivity limit (close to 1) is increased in V4, which means that the fraction of samples for which microphysical retrievals can be attempted is augmented. Secondly, the larger V4 $\beta_{eff}12/k$ yield smaller $D_e12/k$. These two main changes are detailed and quantified in the following sub-sections.

**3.3.1 Fraction of samples in sensitivity range**

Figures 6a and 6b show fractions of samples for which $\beta_{eff}12/10$ and $\beta_{eff}12/08$ are larger than their respective theoretical lower

ranges, which were derived for $D_e = 120\,\mu m$ using the V4 SCO LUT, and in practice are close to 1. For both $\beta_{eff}12/10$ and $\beta_{eff}12/08$, V4 retrievals are possible more than 80 % of the time for $\varepsilon_{eff,12}$ between 0.05 and 0.80 (or about $0.1 - 3.2$ in terms of $\tau_{vis}$). In contrast, the $\varepsilon_{eff,12}$ 80 % range in V3_comp was only $0.15 - 0.7$ for the 12/10 pair and only $0.25 - 0.7$ for the 12/08 pair. As $\varepsilon_{eff,12}$ increases from 0.8 to 0.95 ($\tau_{vis} \sim 6$), which corresponds to clouds that are opaque to CALIOP (see Fig. 2), the $\beta_{eff}12/k$ indices decrease and approach the sensitivity limit, and the fraction of possible retrievals in opaque clouds decreases. This fraction is

notably increased in V4, and is larger at night than for daytime data, reflecting the impact of the cloud radiative temperature corrections introduced in V4. As in V3, this fraction remains lower for the 12/08 pair. One hypothesis is that cloud heterogeneities in dense clouds could induce a larger low bias in the 12/08 pair than in the 12/10 pair (Fauchez et al., 2015). The V4 nighttime retrieval rate is larger than 70 % up to $\varepsilon_{eff,12} = 0.95$ for the 12/10 pair and up to $\varepsilon_{eff,12} = 0.9$ ($\tau_{vis} \sim 4.6$) for the 12/08 pair.

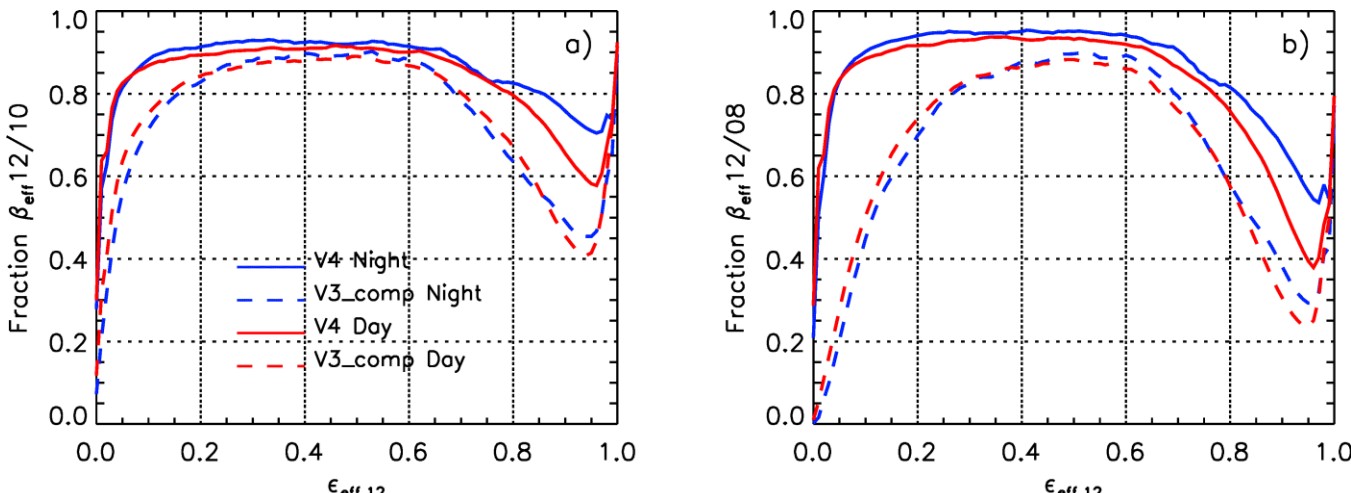

**Figure 6: Fraction of (a) $\beta_{eff}12/10$ and (b) $\beta_{eff}12/08$ values above the effective diameter retrieval sensitivity limit vs. effective emissivity at 12.05 μm in single-layered ice clouds over oceans between 60° S and 60° N in January 2008 in V4 (solid lines) and in V3_comp (dashed lines) during night (blue) and day (red).**

**3.3.2 Changes in effective diameters**

Because the changes in the microphysical indices are larger for the 12/08 pair than for the 12/10 pair, we now assess the changes in the respective diameters, $D_e12/08$ and $D_e12/10$. For meaningful comparisons, the exercise is carried out only for clouds for which both $\beta_{eff}12/10$ and $\beta_{eff}12/08$ are found above the lower sensitivity limit, both in V3 and in V4. The changes in $D_e12/10$ and in $D_e12/08$ are illustrated in Figs. 7a and 7b, respectively. The solid lines represent median $D_e12/k$ derived from V4 $\beta_{eff}12/k$ and the V4 SCO LUT. The dashed lines represent median $D_e12/k$ derived from V3_comp $\beta_{eff}12/k$ and the same V4 SCO LUT, so that

the differences between the solid and the dashed lines are due only to the different microphysical indices. As a result of changes of different amplitude for $D_e12/10$ and $D_e12/08$, the consistency between these two diameters is drastically improved in V4 at $\varepsilon_{eff,12}$ smaller than 0.5. Similar conclusions would be drawn using the V4 CO8 model.

For a complete analysis of the differences between the V3_comp and V4 diameters, the dotted dashed lines show $D_e12/k$ derived using V3_comp and the V3 solid column LUT (Part I), so that the differences between the dotted dashed lines and the dashed lines are due only to the different LUTs. The changes resulting from the LUTs and from the microphysical indices have an opposite effect, regardless of the specific V3 and V4 LUTs chosen for the analysis. As a result, $D_e12/10$ is overall not changed significantly in V4 (solid lines) compared to V3_comp (dotted dashed lines). In contrast, $D_e12/08$ is smaller in V4 by up to 15 µm at $\varepsilon_{eff,12} < 0.2$, because the improved (and increased) $\beta_{eff}12/08$ has the largest impact, and conversely V4 $D_e12/08$ is larger by up to 10 µm at $\varepsilon_{eff,12}$ between 0.2 and 0.9.

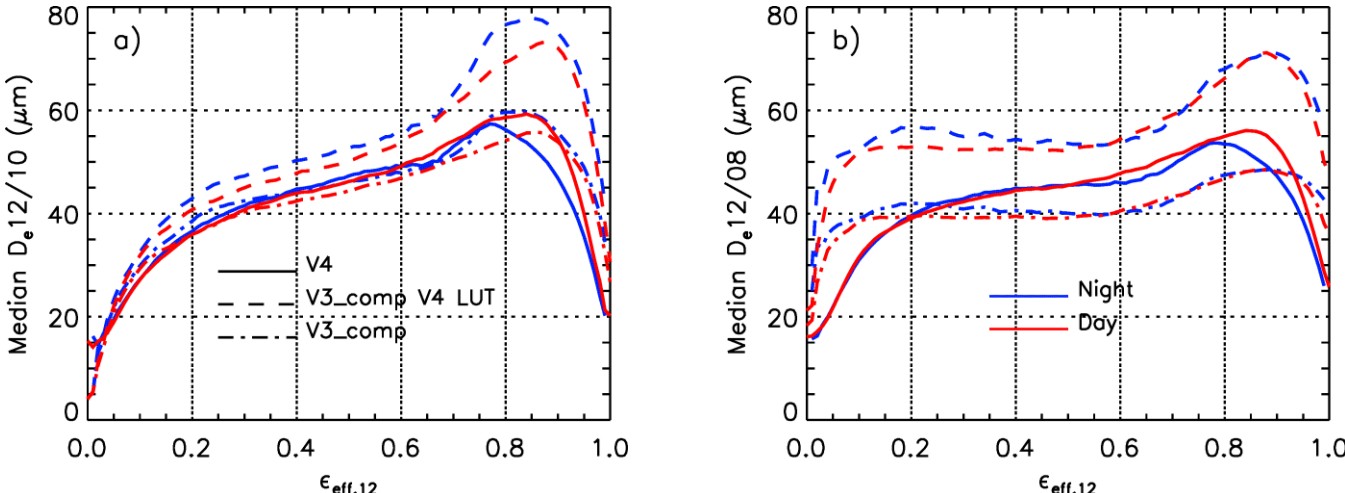

**Figure 7: (a) Median $D_e12/10$ and (b) median $D_e12/08$ vs. effective emissivity at 12.05 µm for the cloud population used in Fig. 6, except that both $\beta_{eff}12/10$ and $\beta_{eff}12/08$ are in the range of possible retrievals, both in V3_comp and in V4. Solid line: V4 with SCO LUT; dashed lines: V3_comp with V4 SCO LUT; dashed dotted line: V3_comp with V3 solid column LUT. Blue: night; red: day.**

### 3.4 V4 microphysical retrievals

We showed in Sect. 3.3 that the fraction of samples with possible microphysical retrievals is significantly increased in V4 (Fig. 6), and that the consistency between the $D_e12/10$ and $D_e12/08$ diameters is drastically improved (Fig. 7). The significant disagreement between $D_e12/10$ and $D_e12/08$ in V3_comp was due to biases of different amplitude in $\beta_{eff}12/10$ and $\beta_{eff}12/08$, and could not be explained by the possible use of an inappropriate ice habit model. Both in V3 and in V4, $D_e$ is retrieved using the ice habit model found in best agreement with IIR in terms of relationship between $\beta_{eff}12/10$ and $\beta_{eff}12/08$. Because the accuracy of IIR $\beta_{eff}12/k$ is improved in V4, the residual discrepancies with respect to the ice habit models are expected to be a genuine piece of information about ice crystal shape. This requires both $\beta_{eff}12/k$ to be found within the sensitivity range, which hereafter will be called "confident" retrievals. Because the population of clouds meeting this requirement is larger in V4 than in V3 and covers a larger range of optical depths, the results in this section will be shown for V4 only.

Theoretically, confident retrievals should be found when $D_e$ is smaller than 120 µm and $\beta_{eff}12/k$ should tend to the upper sensitivity limit for $D_e > 120$ µm. In practice, uncertainties in $\beta_{eff}12/k$ can trigger non-confident retrievals even if $D_e$ is truly smaller than the sensitivity limit, and this is more likely to occur when $D_e$ is close to this limit. Requiring both $\beta_{eff}12/k$ to be in the expected range of values is meant to reinforce the confidence in the retrievals, but doing so implies no systematic bias between both pairs of

channels. This is not exactly true for opaque clouds with $\varepsilon_{eff,12} > \sim 0.8$ (Fig. 6), and consequently the fraction of confident retrievals in opaque clouds is often constrained by the 12/08 pair. Furthermore, the fraction of confident retrievals at large emissivities is larger at night.

### 3.4.1 Effective diameter and ice water path

The histograms of confident $D_e$ and ice water path retrievals (IWP) are shown in Figs. 8a and 8b, respectively, for ST and opaque clouds, and statistics are reported in Table 4. The IWP histograms are computed in logarithmic scale between 0.01 and 1000 g.m$^{-2}$, with log$_{10}$(IWP) bins equal to 0.1. The random uncertainty in $D_e$, noted $\Delta D_e$, is computed based on the LUT selected for the retrieval and the estimated random uncertainty in the $\beta_{eff}12/k$ indices. Median $\Delta D_e/D_e$ values reported in Table 4 are between 34 % and 49 %. The uncertainty in IWP is in large part driven by the uncertainty in $D_e$.

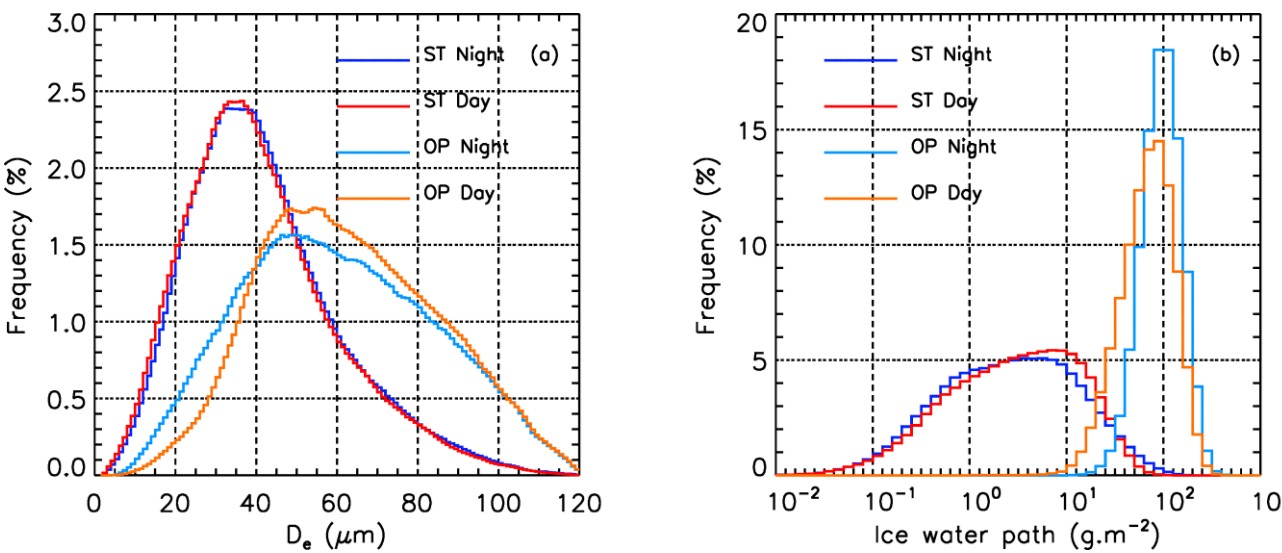

**Figure 8: Histograms of V4 confident retrievals of (a) $D_e$ and (b) Ice water path in single-layered semi-transparent (ST; Night: navy blue; Day: red) and opaque (OP; Night: light blue; Day: orange) ice clouds between 60° S and 60° N over oceans in January 2008.**

Table 4: Statistics associated to V4 effective diameter ($D_e$) and ice water path (IWP) retrievals in single-layered ice clouds between 60° S and 60° N over oceans in January 2008 (see Fig. 8).

| Ice clouds | Semi-transparent | | Opaque | |
|---|---|---|---|---|
| | Night | Day | Night | Day |
| Number of pixels | 167,152 | 201,534 | 98388 | 138,193 |
| Median $\varepsilon_{eff,12}$ | 0.11 | 0.13 | 0.95 | 0.86 |
| Median IIR $\tau_{vis}$ | 0.22 | 0.26 | 5.6 | 3.8 |
| Median $D_e$ (µm) | 39 | 38 | 58.5 | 61 |
| Median $\Delta D_e$ (µm) | 18 | 17 | 28 | 21 |
| Median $\Delta D_e/D_e$ | 0.49 | 0.46 | 0.48 | 0.34 |
| Median IWP (g.m$^{-2}$) | 2.7 | 3.2 | 97 | 71 |
| Median $\Delta$IWP (g.m$^{-2}$) | 1.3 | 1.4 | 50 | 24 |
| Median $\Delta$IWP/IWP | 0.54 | 0.49 | 0.50 | 0.35 |

The ST clouds are optically thin, with median IIR $\tau_{vis}$ of only 0.2 - 0.26. Their nighttime (navy blue) and daytime (red) $D_e$ distributions are nearly identical, with median $D_e$ = 38-39 µm and a peak around $D_e$ = 35 µm. This peak compares well with the mode at 36 µm noted by Dolinar et al. (2019) for single-layered ice clouds with no detectable precipitation as retrieved using the combined CloudSat-CALIPSO 2C-ICE product. IWP (Fig. 8b) is found between 0.03 and 100 g·m$^{-2}$ in ST clouds, with the slightly larger daytime values being explained by the cloud selection and the slightly larger optical depths in the daytime dataset (Table 4). The medium values are around 3 g·m$^{-2}$, with peaks in the distributions at 3 g·m$^{-2}$ and 8 g·m$^{-2}$ for nighttime and daytime data, respectively, and the median relative uncertainty is 50 %. As noted by Berry and Mace (2014), the CloudSat radar is typically insensitive to these thin layers, so that microphysical retrievals in combined CALIPSO-CloudSat products such as 2C-ICE rely on parameterization of the radar reflectivity (Deng et al., 2015) rather than on actual observations. Combining CALIOP and IIR observations appears to be a suitable alternative approach to characterize these thin layers.

The estimated cloud radiative temperature ($T_r$) is at an equivalent altitude located between the CALIOP cloud base and cloud top (Part I). While in case of ST clouds, IIR $D_e$ is a layer average diameter, IIR $D_e$ in opaque clouds is mostly representative of the portion of the cloud seen by CALIOP before the signal is totally attenuated. These opaque clouds have median $\varepsilon_{eff,12}$ equal to 0.95 at night but only 0.86 for daytime data, with median IIR $\tau_{vis}$ equal to 5.6 and 3.8, respectively. Median $D_e$ in opaque clouds is around 60 µm and the distributions peak at 50 µm. It is larger than in ST clouds, which is consistent with retrievals based on AIRS thermal infrared data (Guignard et al., 2012; Kahn et al., 2018). The different nighttime and daytime $D_e$ and IWP distributions in opaque clouds are explained by the different ranges of optical depth and the different amplitudes of the radiative temperature correction (Figs. 2 and 4). In opaque clouds, the retrieved IWP lies between 10 and only 300 g.m$^{-2}$. The upper limit is due to the fact that $D_e$ cannot be larger than 120 µm and because cloud optical depths inferred from IIR effective emissivities saturate and are typically smaller than 15 (Fig. 4).

### 3.4.2 Ice habit model selection

Recall that $D_e$ is retrieved using the habit model (SCO or CO8) that agrees the best with IIR in terms of the relationship between $\beta_{eff}$12/10 and $\beta_{eff}$ 12/08. As seen in Fig. 9, the SCO habit model is selected in 80 % of the ST clouds of $T_r$ < 205 K. This fraction steadily decreases down to 60 % as $T_r$ increases up to 230 K (Fig. 9b) and remains stable above 230 K. This result is qualitatively consistent with previous findings using V3 (Garnier et al., 2015), and, as was discussed in this paper, both the IIR model selection and the mean CALIOP integrated particulate depolarization ratio (in black in Fig. 9b) indicate changes of crystal habit with temperature. In opaque clouds (Fig. 10), both the IIR model selection and the CALIOP depolarization ratio between 200 K and 230 K are less temperature-dependent than in ST clouds. The difference between mean $D_e$12/10 and mean $D_e$12/08 in black and grey in Figs. 9c and 10c is a measure of the residual mismatch between IIR observations and the selected model. We see two temperature regimes, that is, below and above 225 K, with a better agreement between IIR and the LUTs at the warmer temperatures. This suggests that the V4 models are better suited for warmer clouds and that they do not perfectly reproduce the infrared spectral signatures of colder clouds composed of small crystals. It is acknowledged that the highly variable ice particle shapes found in ice clouds (Lawson et al., 2019 and references therein) are likely not fully reproduced through the two models chosen for the V4 algorithm. It is further noted that the Clouds and the Earth's Radiant Energy System (CERES) science team is planning to use a two-habit model for retrievals in the visible/near infrared spectral domain (Liu et al., 2014; Loeb et al., 2018). This model would be a mixture of two habits (single column and an ensemble of aggregates) whose mixing ratio would vary with ice crystal maximum dimension, with single columns prevailing for the smaller dimensions. Interestingly, our findings appear to be consistent with this approach.

In both thin ST clouds (Fig. 9c) and opaque clouds (Fig. 10c), $D_e$ increases with cloud radiative temperature until it reaches a maximum value around 250 K in ST clouds and 230 K in opaque clouds. Kahn et al. (2018) found that for clouds of emissivity smaller than 0.98 (or $\tau_{vis}$ smaller than about 8), $D_e$ is maximum and around 50 μm at 230 K, which is consistent with our findings, keeping in mind that clouds with emissivity smaller than 0.98 are found in both our ST and opaque clouds. The increase of cloud
average $D_e$ with cloud radiative temperature in ST clouds (Fig. 9c) is in general agreement with numerous previous findings (e.g. Hong and Liu, 2015). The decrease of $D_e$ between $T_r = 250$ K and 260 K for ST clouds is possibly due to an increasing fraction of small liquid droplets in these prevailingly ice layers, which would be consistent with the fact that CALIOP integrated particulate depolarization ratio decreases from 0.37 to 0.30 (Fig. 9b). Similar comments apply for opaque clouds for $T_r$ between 230 and 260 K. Using combined POLDER (POLarization and Directionality of the Earth's Reflectances) and MODIS data, Van Diedenhoven
et al. (2020) found that $D_e$ at the top of thick clouds of optical depth larger than 5 is maximum at cloud top temperature equal to 250 K, rather than $T_r = 230$ K for opaque clouds. This discrepancy might be partly explained if the cloud radiative altitude is higher in the cloud than the cloud top derived from the visible observations, which could also explain that $D_e$ shown in Van Diedenhoven et al. (2020) is larger than in this study.

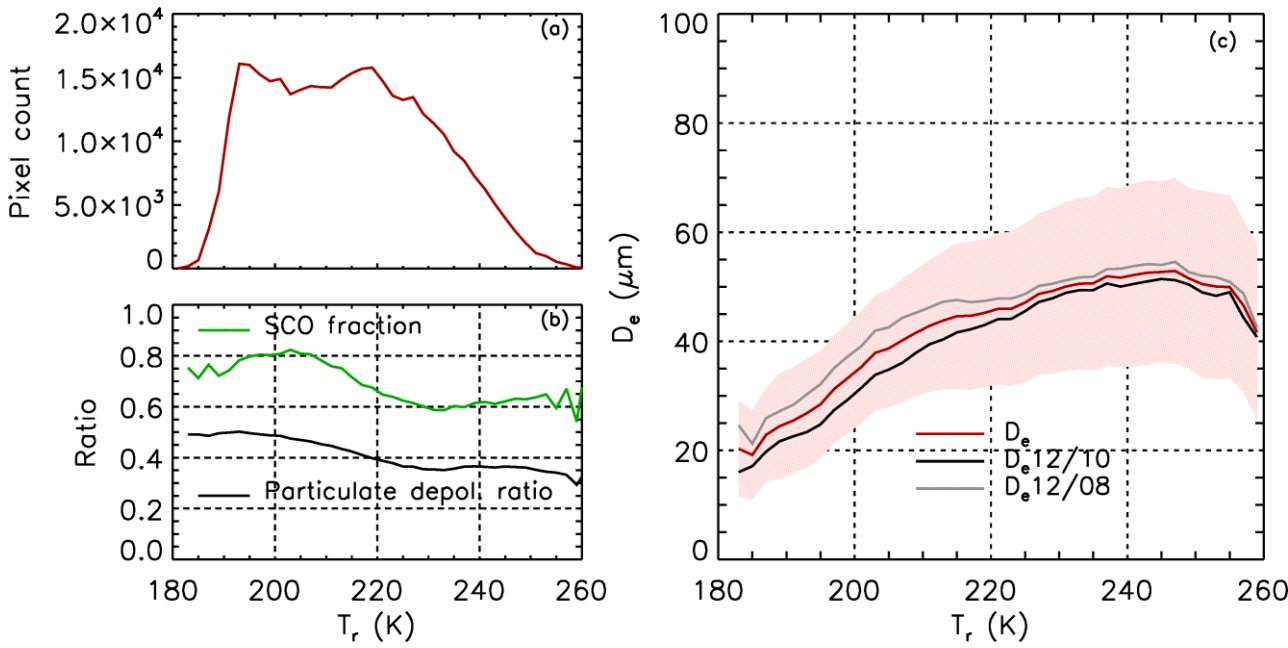

**Figure 9: IIR V4 confident retrievals vs. radiative temperature in semi-transparent ice clouds over oceans between 60°S and 60°N in January 2008. (a) Pixel count; (b) fraction of retrievals using the SCO model (green) and mean CALIOP integrated particulate depolarization ratio (black); (c) mean $D_e$ (red) ± mean absolute deviation (shaded area), mean De12/10 (black) and mean De12/08 (grey).**

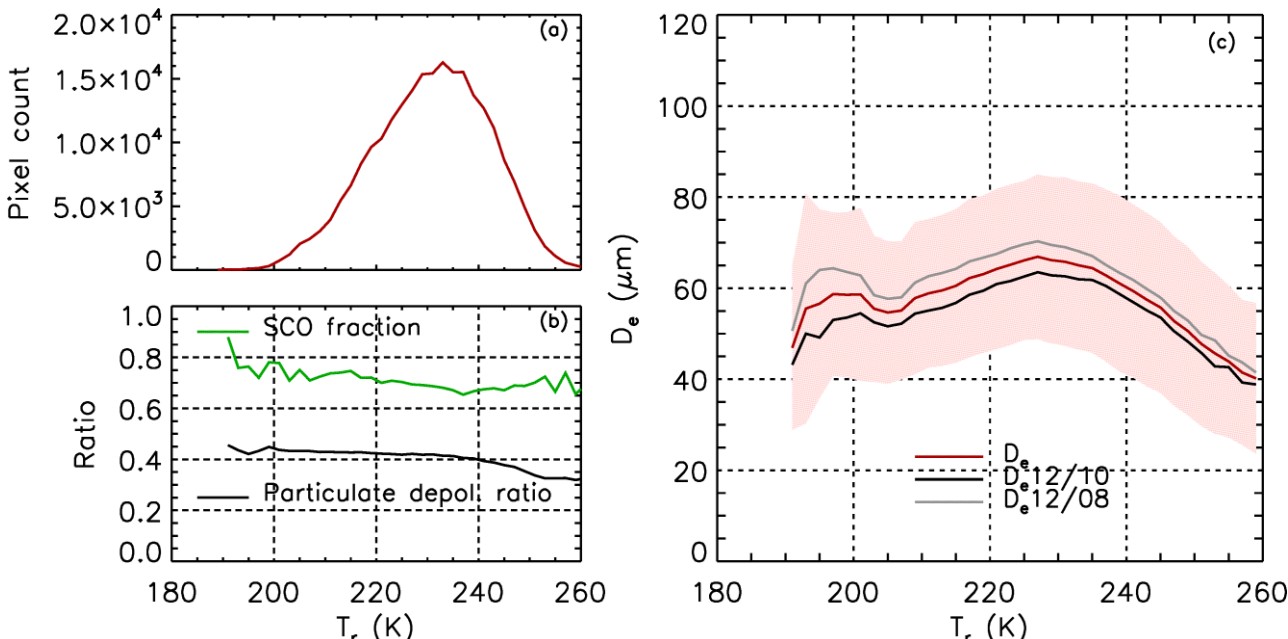

**Figure 10: Same as Fig. 9, but for opaque clouds.**

### 3.4.3 Retrievals using parameterizations from in situ formulation

The IIR algorithm takes advantage of the relationship between $\beta_{eff}12/10$ and $\beta_{eff}12/08$ to identify the ice habit model that best
matches the observations and thereby provide information about both ice crystal shape and effective diameter. Another approach
would be to use only $\beta_{eff}12/10$ and prescribed LUTs. This approach was adopted by Mitchell et al. (2018), who derived four sets
of LUTs using extensive in situ measurements rather than pure modeling. In Part I, we compared these four sets of $\text{ß}_{eff}12/10 - D_e$
relationships with the relationships derived from the V4 SCO and CO8 models. The four sets of $D_e$ derived from $\text{ß}_{eff}12/10$ using
this independent approach are reported in the IIR product for the user's convenience. Figure 11 compares $D_e$ computed by the
analytic function derived by Mitchell et al. (2018) with $D_e12/10$ from the CO8 and the SCO models. Relationships derived from
the SPARTICUS (blue) and the TC4 (red) field campaign were computed in two ways: by setting the first bin of the measured
particle size distribution (PSD) (D < 15 µm) to 0 (i.e. $N(D)_1 = 0$, dashed lines) and without modifying the distribution (i.e. $N(D)_1$
unmodified, solid lines). As discussed in Part I, the differences between the six sets of retrievals illustrate the possible impacts of
the LUTs and of the PSDs. Because the presence of small particles in the unmodified PSD causes $\beta_{eff}12/10$ to increase faster than
$D_e$, assuming $N(D)_1 = 0$ yields smaller values of $D_e$ for a given $\beta_{eff}12/10$ than when $N(D)_1$ is not modified. Even though this was
not the original intent, comparing median $D_e$ with or without setting $N(D)_1$ to 0 also illustrates the impact of possible vertical
inhomogeneities of $D_e$ within the cloud layer (Zhang et al., 2010). Nevertheless, the overall impact of vertical variations on
$\beta_{eff}12/10$ also depends on the in-cloud IIR weighting function, which is related to the cloud extinction profile (Part I). In Mitchell
et al. (2020), the mean $D_e$ calculated from the SPARTICUS unmodified $\text{ß}_{eff}12/10 - D_e$ relationship (applied at mid-latitudes) and
the TC4 $N(D)_1 = 0$ $\text{ß}_{eff}12/10 - D_e$ relationship (applied in the tropics) was compared against the in situ climatology of mean volume
radius, $R_v$, reported in Krämer et al. (2020) after converting $D_e$ to $R_v$. The retrieved $R_v$ tended to be no more than ~ 20% smaller
than the in situ $R_v$ for temperatures between 208 and 233 K.

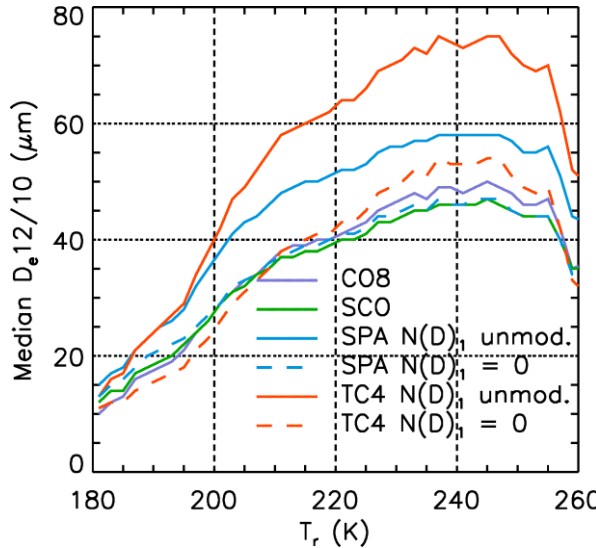

**Figure 11: Median $D_e12/10$ from the V4 CO8 (purple) and SCO (green) models, and from analytical functions derived by Mitchell et al. (2018) during the SPARTICUS (blue) and TC4 (red) field experiments using $N(D)_1$ unmodified (solid) or $N(D)_1 = 0$ (dashed). Same dataset as in Fig. 9.**

### 3.5 Comparisons with MODIS

Figure 12 compares IIR confident retrievals and co-located Aqua/MODIS Collection 6 daytime retrievals from the visible/2.1 µm
and visible/3.7 µm pairs of channels (Platnick et al., 2017, and references therein) in single-layered clouds classified as high
confidence ROI by CALIOP and as ice clouds by MODIS. MODIS $\tau_{vis}$ and $D_e$ at 1-km resolution are from the MYD06 product
and co-location with CALIPSO is from the AERIS/ICARE CALTRACK product. Analyses are over oceans between 30° S and
30° N in January 2008 separately for CALIPSO ST and opaque clouds. Figures 12a and 12b show the population of clouds with
IIR retrievals (black), with MODIS retrievals at both 2.1 and 3.7 µm (brown), and with both IIR and MODIS retrievals (orange)
for which comparisons in Figs. 12c-f are shown. Figures 12a and 12b characterize these cloud populations as a function of IIR $T_r$
and IIR $\varepsilon_{eff,12}$, respectively. For ST clouds (thin lines), the IIR-MODIS comparisons are constrained by the availability of MODIS
retrievals, and the compared ST clouds have $\varepsilon_{ff,12}$ typically larger than 0.2 (Fig. 12b). In contrast, comparisons in opaque clouds
(thick lines) are limited by the availability of IIR retrievals. Figures 12c and 12e show median $D_e$ from IIR (red), MODIS 2.1
(green) and MODIS 3.7 (blue) vs. $T_r$ for ST (Fig. 12c) and opaque (Fig. 12e) clouds. The vertical lines are between the 25[th] and
495 75[th] percentiles. Similarly, Figs. 12d and 12f show the corresponding $\tau_{vis}$ values. Only one MODIS $\tau_{vis}$ is shown because the
retrievals from both pairs of MODIS channels are nearly identical.

For ST clouds, MODIS 2.1 $D_e$ is larger than IIR by 15 µm on average. IIR and MODIS 3.7 $D_e$ are in good agreement for $T_r < 205$
K where $D_e$ is $< 40$ µm and IIR $\tau_{vis}$ is $< 0.5$, and they progressively depart from each other as $T_r$ increases and MODIS 3.7 increases
and approaches MODIS 2.1. MODIS $\tau_{vis}$ is larger than IIR by 0.3 to 0.2. This small but systematic bias is not seen when comparing
CALIOP and IIR (not shown). The MODIS 2.1 $D_e$ – $T_r$ relationships are similar for ST and opaque clouds, which is not the case
for MODIS 3.7 and IIR. For opaque clouds, IIR $D_e$. is larger than in ST clouds and is in good agreement with MODIS 2.1 at $T_r <$
225 K. MODIS 3.7 $D_e$ exhibits a similar increase with temperature as seen with the two other data sets, but it is shifted by -10 µm.
At $T_r > 225$ K, MODIS $D_e$ 2.1 continues to increase up to 100 µm at 255 K, whereas IIR remains stable around 60 µm and MODIS
3.7 increases slowly to approach the same plateau as IIR around 60 µm. As seen in Fig. 12f, both MODIS and IIR indicate moderate
optical depths in these opaque clouds where comparisons are possible, with median values ranging between 2.5 and 6 at $T_r < 250$
K, IIR being smaller than MODIS by about 0.4.

Kahn et al. (2015) found that MODIS 2.1 $D_e$ is typically larger than AIRS $D_e$ by 10-20 µm, and that MODIS 3.7 is in better agreement with AIRS on average. These results, which were for clouds of optical depth between 0.5 and 2 over oceans, are consistent with our findings for ST clouds. The MODIS and IIR techniques exhibit different non-linear sensitivities to particle size,

so that vertical inhomogeneities of the effective diameter can yield three different retrieved $D_e$ (Zhang et al., 2010). This could explain that IIR $D_e$ is found in better agreement with MODIS 3.7 in ST clouds while MODIS 2.1 is clearly larger (Zhang et al., 2010). At $T_r > 220$ K, IIR $D_e$ is around 50-60 µm and smaller than both MODIS 2.1 and 3.7. We note that the agreement with MODIS would be improved using the parameterized functions derived from the unmodified in-situ PSDs that were presented in Sect. 3.4.3, but that the modified PSDs would yield similar results. For clouds of moderate optical depth as found in our population

of opaque clouds, MODIS 3.7 is very sensitive to cloud top while MODIS 2.1 senses deeper into the cloud (Zhang et al., 2010; Platnick, 2000), and the smaller MODIS 3.7 $D_e$ as observed in Fig. 12e suggests that the effective diameter is smaller at cloud top than deeper into the cloud. IIR $D_e$ might be larger than MODIS 3.7 and in better agreement with MODIS 2.1 for opaque clouds at $T_r < 220$ K because the IIR weighting function is deeper into the cloud than at 3.7 µm, which is agreement with simulations by Zhang et al. (2010). In conclusion, distinct sensitivity to possible cloud vertical and horizontal (Fauchez et al., 2018) inhomogeneity

likely contributes to the observed differences.

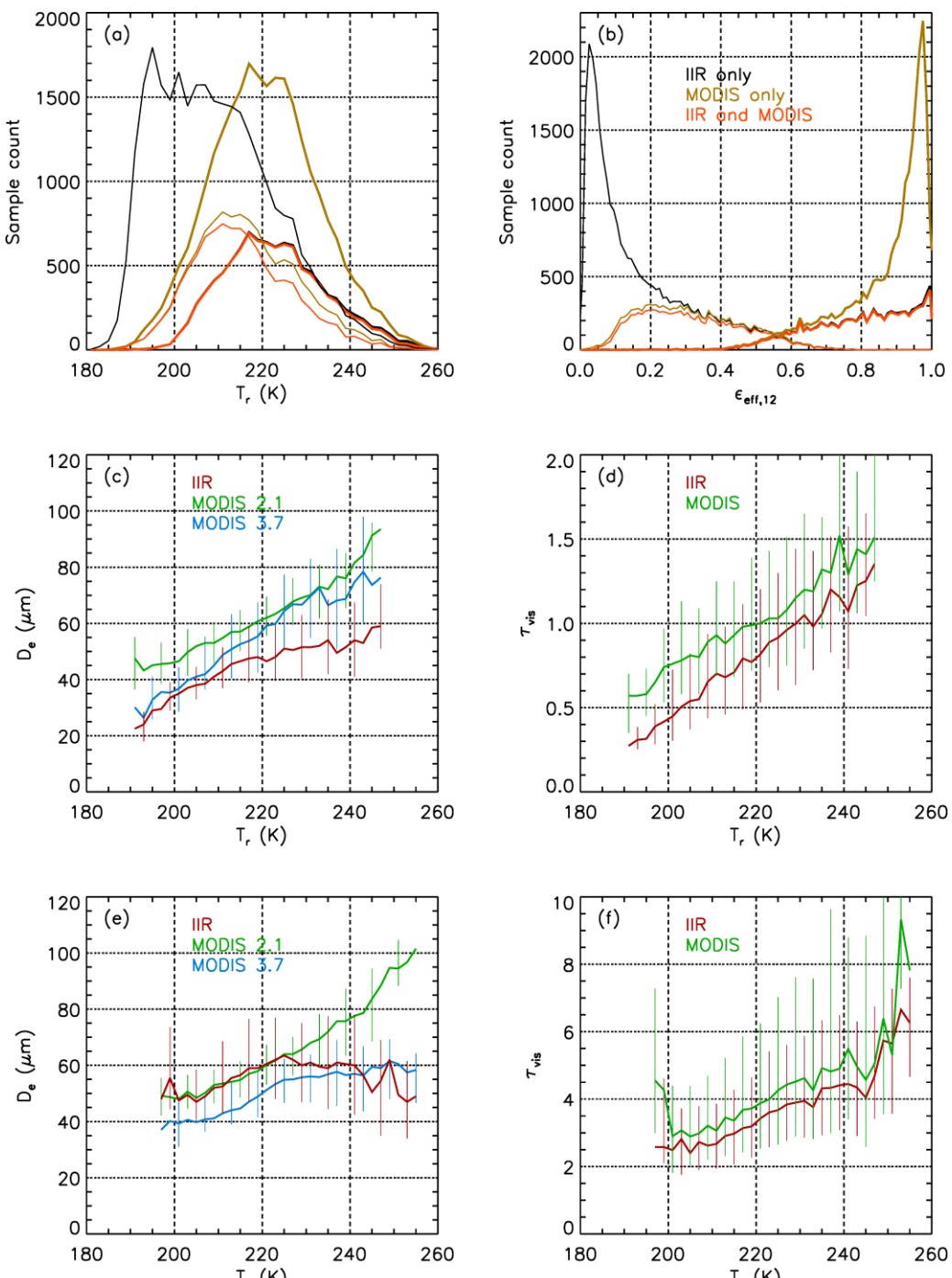

**Figure 12: IIR and MODIS comparisons over oceans between 30° S and 30° N in January 2008 for single-layered high confidence ROI clouds with MODIS ice phase. Top: distributions of (a) IIR radiative temperature and (b) IIR effective emissivity at 12.05 μm in ST (thin lines) and opaque (thick lines) clouds where IIR has confident retrievals (black), MODIS has successful retrievals at 2.1 and 3.7 μm (brown), and both IIR and MODIS retrievals are successful and can be compared (orange). Median $D_e$ vs. $T_r$ from IIR (red), MODIS 2.1 (green) and MODIS 3.7 (blue) in ST (c) and opaque (e) clouds; Median $\tau_{vis}$ from IIR (red) and MODIS (green) in ST (d) and opaque (f) clouds. The vertical bars in panels (c-f) are between the 25th and 75th percentiles.**

## 4 V4 Retrievals in liquid water clouds

The only difference between effective emissivity V4 retrievals in liquid and ice clouds is that $T_r$ is taken as the temperature at the CALIOP centroid altitude ($T_c$) in case of liquid water clouds, whereas this initial temperature estimate is further corrected in case

of ice clouds. It is recalled that $D_e$ of liquid droplets are retrieved using the water LUTs (Part I) and that liquid water path is derived from $D_e$ and $\varepsilon_{eff,12}$ (Eq. 10 in Part I).

Following a similar approach as for ice clouds, the results are shown for scenes over oceans between 60° S and 60° N that contain one single cloud layer classified as high confidence water by the CALIOP phase algorithm. Because liquid water clouds are statistically warmer than ice clouds, the radiative contrast is typically smaller than for ice clouds. Because uncertainties are inversely proportional to this radiative contrast (Part I), they increase very rapidly when the radiative temperature contrast, that is the difference between the clear air TOA background brightness temperature and the TOA blackbody brightness temperature, is smaller than 10 K. In order to prevent very large uncertainties associated with very small radiative contrast, the results are presented for clouds in the free troposphere with centroid altitude above 4 km. For this cloud population, the radiative temperature contrast is larger than 10 K, and it increases on average from 15 K at 4 km to 50 K at 10 km where the highest water clouds are found (not shown). Most of these sampled liquid clouds are composed of supercooled droplets.

Our results are presented in Sect. 4.1 to 4.3 and comparisons with MODIS are shown in Sect. 4.4.

### 4.1 Effective emissivity in channel 12.05

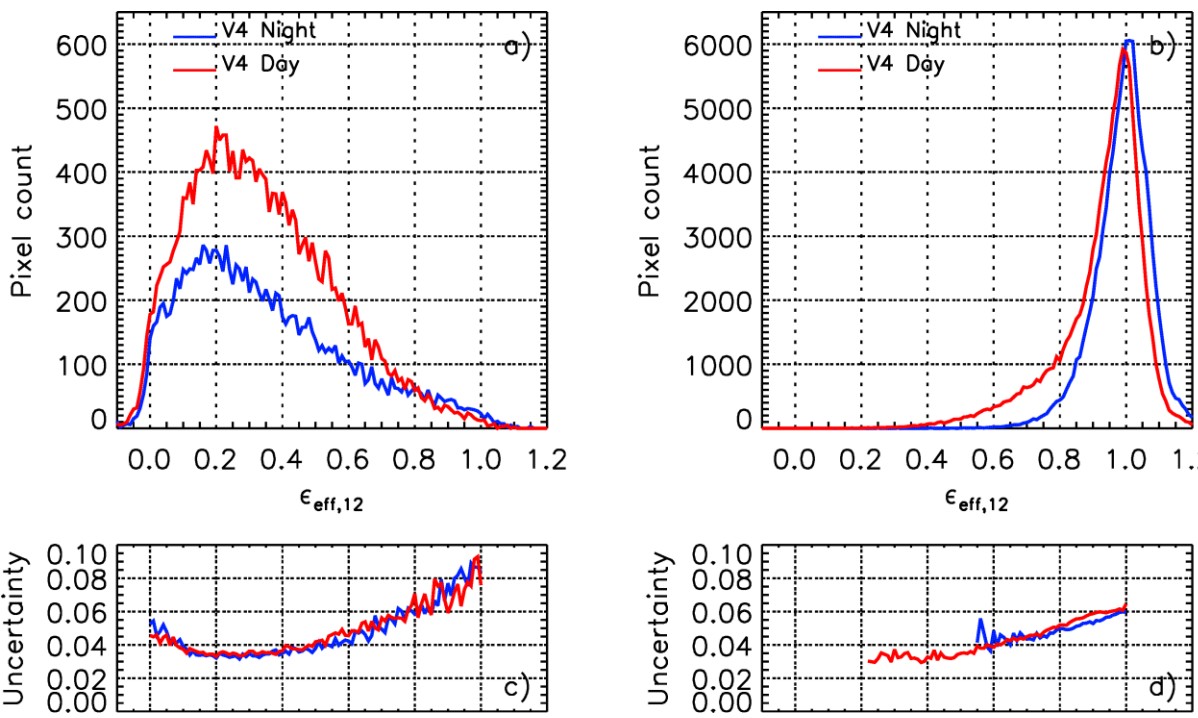

**Figure 13: V4 effective emissivity distribution at 12.05 µm in (a) ST and (b) opaque single-layered liquid water clouds of centroid altitude > 4 km over oceans between 60° S and 60° N in January 2008 for nighttime (blue) and daytime (red) data. Panels (c) and (d) are the V4 median random uncertainties corresponding to panels (a) and (b), respectively.**

Figures 13a and 13b show the distributions of V4 $\varepsilon_{eff,12}$ in ST and opaque liquid water clouds, respectively, for the month of January 2008 between 60° S and 60° N over ocean, for clouds with centroid altitude > 4 km. Figures 13c and 13d show the respective median random uncertainties., which are about twice as large as the uncertainties in ice clouds (Figs. 2c and 2d) because of the smaller radiative contrast. Only 17 % of these clouds are ST (Figs. 13a and 13c). Unlike in ST ice clouds, the distributions peak at $\varepsilon_{eff,12} \sim 0.2$, and non-physical negative emissivity values are found in only 2 % of the pixels. The $\varepsilon_{eff,12}$ distributions in opaque clouds peak at 1.02 at night and at 0.99 for daytime data, with an estimated uncertainty of ± 0.06. The spread around these peaks

is larger than for ice clouds, which is explained by the larger uncertainties and specifically to a larger sensitivity to a wrong estimate of $T_r$. Thus, the nighttime and daytime fractions of samples with $\varepsilon_{eff,12} > 1$, for which no microphysical retrievals are possible, are 45 % and 27 %, respectively. The daytime distributions in opaque clouds exhibit a tail down to $\varepsilon_{eff,12} \sim 0.4$, while at night, the lowest $\varepsilon_{eff,12}$ is $\sim 0.65$, which is very similar to what was observed for opaque ice clouds (Fig. 2b). This similarity suggests that emissivity retrievals in ice and liquid water clouds are consistent, notwithstanding the unavoidable larger uncertainties in the latter ones.

**4.2 Inter-channel effective emissivity differences**

The variations with $\varepsilon_{eff,12}$ of the V4 $\Delta\varepsilon_{eff}$12-k inter-channel effective emissivity differences for the 12-10 and 12-08 pairs are shown in Figs. 14a and 14b, respectively. The nighttime (blue) and daytime (red) curves are median values, and the shaded gray areas are between the V4 nighttime 25th and 75th percentiles. As for ice clouds, both $\Delta\varepsilon_{eff}$12-k tend nicely to 0 at $\varepsilon_{eff,12} \sim 0$, owing to the improved computed background radiances demonstrated previously, which has a beneficial effect on retrievals in any ST layer. Both $\Delta\varepsilon_{eff}$12-k have a second minimum at $\varepsilon_{eff,12} \sim 1$, as expected, and this minimum is found slightly larger than 0. Both $\Delta\varepsilon_{eff}$12-k and therefore both $\beta_{eff}$12/k are notably larger than for ice clouds (see Fig. 3), reflecting the presence of smaller particles in the liquid water distributions (Giraud et al., 2001; Mitchell and d'Entremont, 2012). As shown by Avery et al. (2020), the IIR microphysical indices are unambiguously larger in clouds classified as liquid water by the CALIOP phase algorithm than in clouds classified as ice.

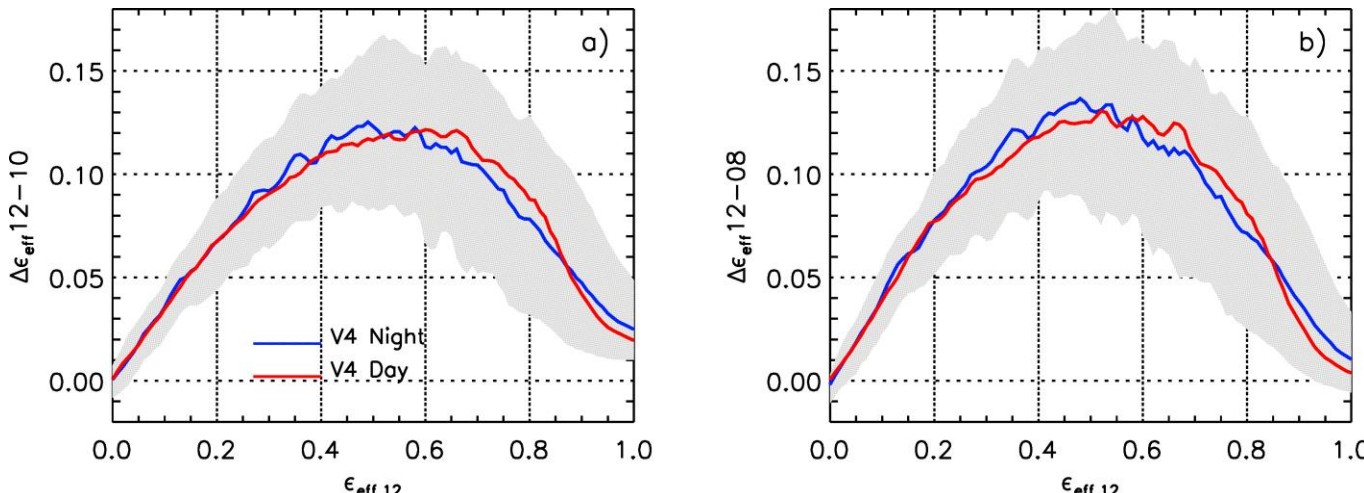

**Figure 14: V4 IIR inter-channel (a) $\Delta\varepsilon_{eff}$12-10 and (b) $\Delta\varepsilon_{eff}$12-08 effective emissivity differences vs. effective emissivity at 12.05 μm in single-layered liquid water clouds of centroid altitude > 4 km over oceans between 60° S and 60° N in January 2008. The blue and red curves are median values for nighttime and daytime data, respectively. The shaded gray areas are between the V4 nighttime 25th and 75th percentiles.**

**4.3 Microphysical retrievals**

**4.3.1 Effective diameter and liquid water path**

As previously, retrievals are deemed confident when both $\beta_{eff}$12/k are found within the sensitivity range, which corresponds to $D_e$ = 60 μm for liquid clouds. The fraction of confident retrievals is found similar in liquid water clouds of centroid altitude > 4 km and in ice clouds. Following the same presentation as for ice clouds, the histograms of confident $D_e$ and liquid water path retrievals (LWP) are shown in Figs. 15a and 15b, respectively, for ST and opaque clouds, and statistics are reported in Table 5.

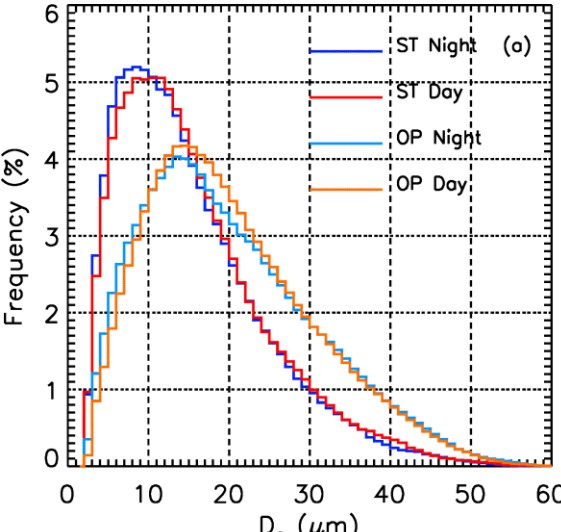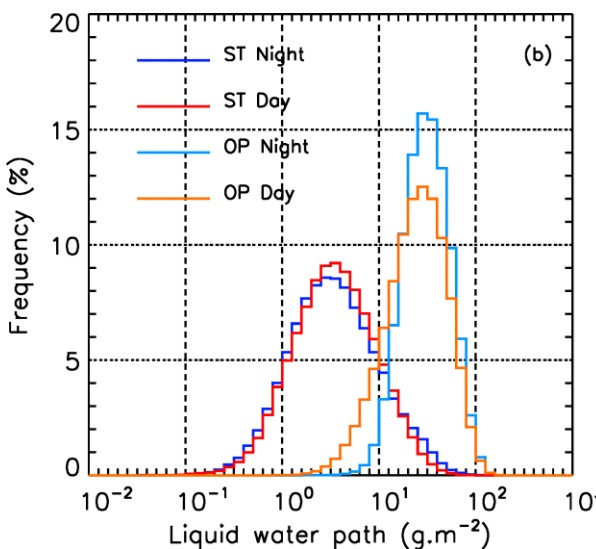

**Figure 15: Histograms of V4 confident retrievals of (a) $D_e$ and (b) liquid water path in single-layered semi-transparent (ST; Night: navy blue; Day: red) and opaque (OP; Night: light blue; Day: orange) liquid water clouds of centroid altitude > 4 km between 60° S and 60° N over oceans in January 2008.**

Note that the IIR retrievals shown in Fig. 15 are for a population of optically thin water clouds: median $\tau_{vis}$ is only 0.9 in ST clouds and between 4 and 5 in opaque clouds. Both in ST and in opaque clouds, the nighttime and daytime $D_e$ histograms are similar. In ST clouds, median $D_e$ is 13 µm and median liquid water path is 3.4 g.m$^{-2}$ with a median random uncertainty of 1.2 g.m$^{-2}$. In opaque clouds, median $D_e$ is 18 µm and median liquid water path is 25-31 g.m$^{-2}$ with a median random uncertainty of 10-15 g.m$^{-2}$. The

maximum retrieved LWP is about 100 g.m$^{-2}$, consistent with the infrared saturation range of 40-60 g.m$^{-2}$ reported by Marke et al. (2016) who combined microwave and infrared ground-based observations to improve LWP and $D_e$ retrievals in "thin" clouds that they defined as LWP < 100 g.m$^{-2}$. The authors report $D_e$ between 10 and 14 µm in "thin" clouds of top altitude < ~ 1 km, which agrees well with the peaks of our distributions.

Table 5: Statistics associated to V4 effective diameter ($D_e$) and liquid water path (LWP) retrievals in single-layered liquid water clouds of centroid altitude > 4 km between 60° S and 60° N over oceans in January 2008 (see Fig. 15).

| Ice clouds | Semi-transparent | | Opaque | |
|---|---|---|---|---|
| | Night | Day | Night | Day |
| Number of pixels | 11,562 | 18,887 | 36,998 | 54,169 |
| Median $\varepsilon_{eff,12}$ | 0.33 | 0.34 | 0.94 | 0.89 |
| Median IIR $\tau_{vis}$ | 0.88 | 0.87 | 5.23 | 4.15 |
| Median $D_e$ (µm) | 13 | 13.5 | 18 | 18.5 |
| Median $\Delta D_e$ (µm) | 5.6 | 5.7 | 9 | 8 |
| Median $\Delta D_e/D_e$ | 0.46 | 0.45 | 0.52 | 0.42 |
| Median LWP (g.m$^{-2}$) | 3.3 | 3.4 | 31 | 25 |
| Median $\Delta$LWP (g.m$^{-2}$) | 1.1 | 1.2 | 15 | 10 |
| Median $\Delta$LWP/LWP | 0.35 | 0.35 | 0.48 | 0.39 |

### 4.3.2 Analyses vs. radiative temperature

IIR retrievals in ST liquid water clouds are shown in Fig. 16 as a function of $T_r$, highlighting that most of these liquid clouds of
centroid altitude > 4 km are supercooled, with $T_r$ ranging between 235 and 280 K (Fig. 16a). Mean IIR $D_e$ (Fig. 16b, red) increases
steadily from 11 µm at 242 K to 18 µm at 270 K, while mean CALIOP particulate depolarization ratio (Fig. 16c) is constant and
around 0.1. These thin clouds are likely radiation-driven, and the increase of layer average $D_e$ with layer radiative temperature
could indicate growth through vapor deposition. In addition, there is an increasing probability for supercooled droplets to freeze
as temperature decreases. As $T_r$ decreases from 242 K to 235 K, the number of samples drops quickly, $D_e$ increases up to 24 µm,
and CALIOP depolarization ratio increases very significantly, confirming a rapid transition to ice phase. At $T_r$ > 270 K, $D_e$
continues to increase slightly up to 20 µm, while CALIOP integrated particulate depolarization ratio decreases. As seen in Fig.
16b, $D_e$12/10 and $D_e$12/08 are in fair agreement. The mean $D_e$12/10-$D_e$12/08 difference increases from -2 µm at 275 K to + 3 µm
at 245 K. This slight temperature-dependent discrepancy between the IIR observations and the water LUT could be explained by
the fact that the complex refractive index is temperature dependent, as reported by Zasetsky et al. (2005) and Wagner et al. (2005),
the complex refractive index of supercooled water being intermediate between warm water and ice (Rowe et al., 2013). Further
investigations will be carried out to establish whether the residual discrepancy between $D_e$12/10 and $D_e$12/08 would be reduced by
using a new set of temperature-dependent indices, following the approach in Rowe et al. (2013). Nevertheless, these simple
observations give confidence in the new V4 IIR $D_e$ retrievals in ST liquid clouds.

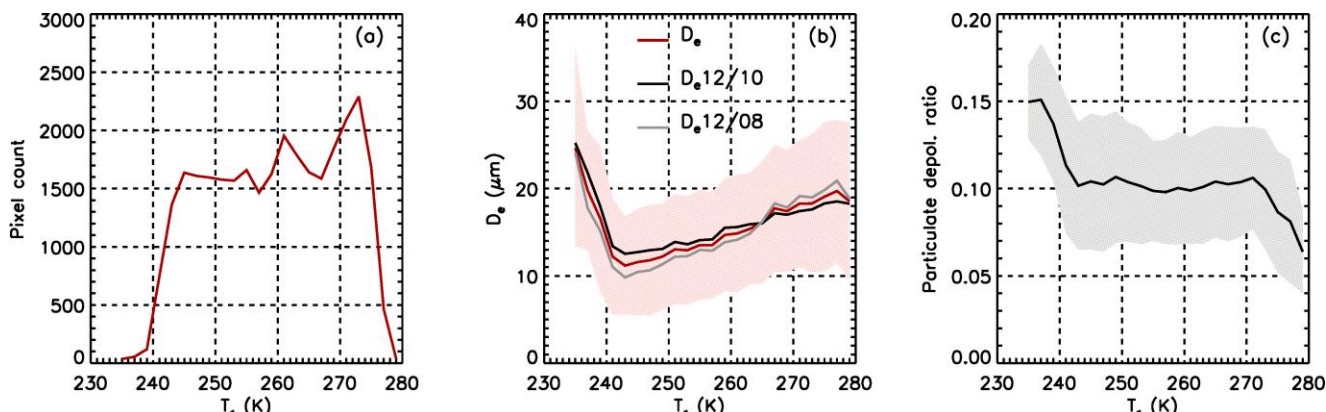

**Figure 16: IIR confident retrievals vs. radiative temperature in ST liquid water clouds of centroid altitude > 4 km over oceans between
60° S and 60° N in January 2008. (a) Pixel count; (b) mean $D_e$ (red), $D_e$12/10 (black) and $D_e$12/08 (grey); (c) mean CALIOP integrated
particulate depolarization ratio. The shaded areas in panels (b) and (c) represent mean ± mean absolute deviation.**

### 4.4 Comparisons with MODIS

IIR confident retrievals in liquid water clouds were compared with MODIS Collection 6 retrievals from the visible/2.1 µm and
visible/3.7 µm pairs of channels for clouds also classified as liquid water by MODIS. The results are shown in Fig. 17, following
the same presentation as in Fig. 12 for ice clouds. Again, cloud centroid altitude is chosen to be higher than 4 km, and, as previously
for ice clouds, the comparisons shown in Figs. 17 c-f are limited to those pixels for which the IIR, MODIS 2.1 and MODIS 3.7
retrievals (orange curves in Figs. 17a and 17b) were all successful. As seen in Fig. 17a, $T_r$ spans between 235 K and 280 K, and
most of these sampled clouds are composed of supercooled droplets. In ST clouds, the three datasets show an increase of median
$D_e$ (Fig. 17c) as $T_r$ increases from 243 K to 270 K, but with different slopes: IIR $D_e$ increases with $T_r$ from 10 to 20 µm whereas
both MODIS 2.1 and 3.7 are larger than about 20 µm, and the differences between IIR and MODIS decrease as temperature
increases. As seen in Fig. 17d, these supercooled water clouds have optical depths between 1.5 and 2 according to MODIS, whereas
IIR $\tau_{vis}$ is 30 to 40 % smaller. In contrast, the three sets of $D_e$ exhibit similar variations with $T_r$ in opaque clouds (Fig. 17e). IIR $D_e$

(red) is systematically smaller than MODIS 2.1 (green), by 8 µm on average. This is fairly consistent with findings by Di Noia et al. (2019) who compared MODIS 2.1 with new retrievals from POLDER-3 measurements, and found that MODIS 2.1 effective radius was larger by about 3 µm ($D_e$ larger by 6 µm) for high oceanic clouds having pressures lower than 600 hPa. MODIS 3.7 retrievals (blue) are weighted closer to the top of the cloud than the corresponding MODIS 2.1 retrievals (Platnick, 2000), and are larger than IIR $D_e$ estimates by only 3 µm. This is encouraging, despite of the seemingly temperature-dependent discrepancy

between MODIS and IIR $\tau_{vis}$ (Fig. 17f), where median IIR $\tau_{vis}$ (red) saturates around $\tau_{vis} = 5$ while median MODIS increases up to 15 at 240 K. More work is necessary to understand these differences.

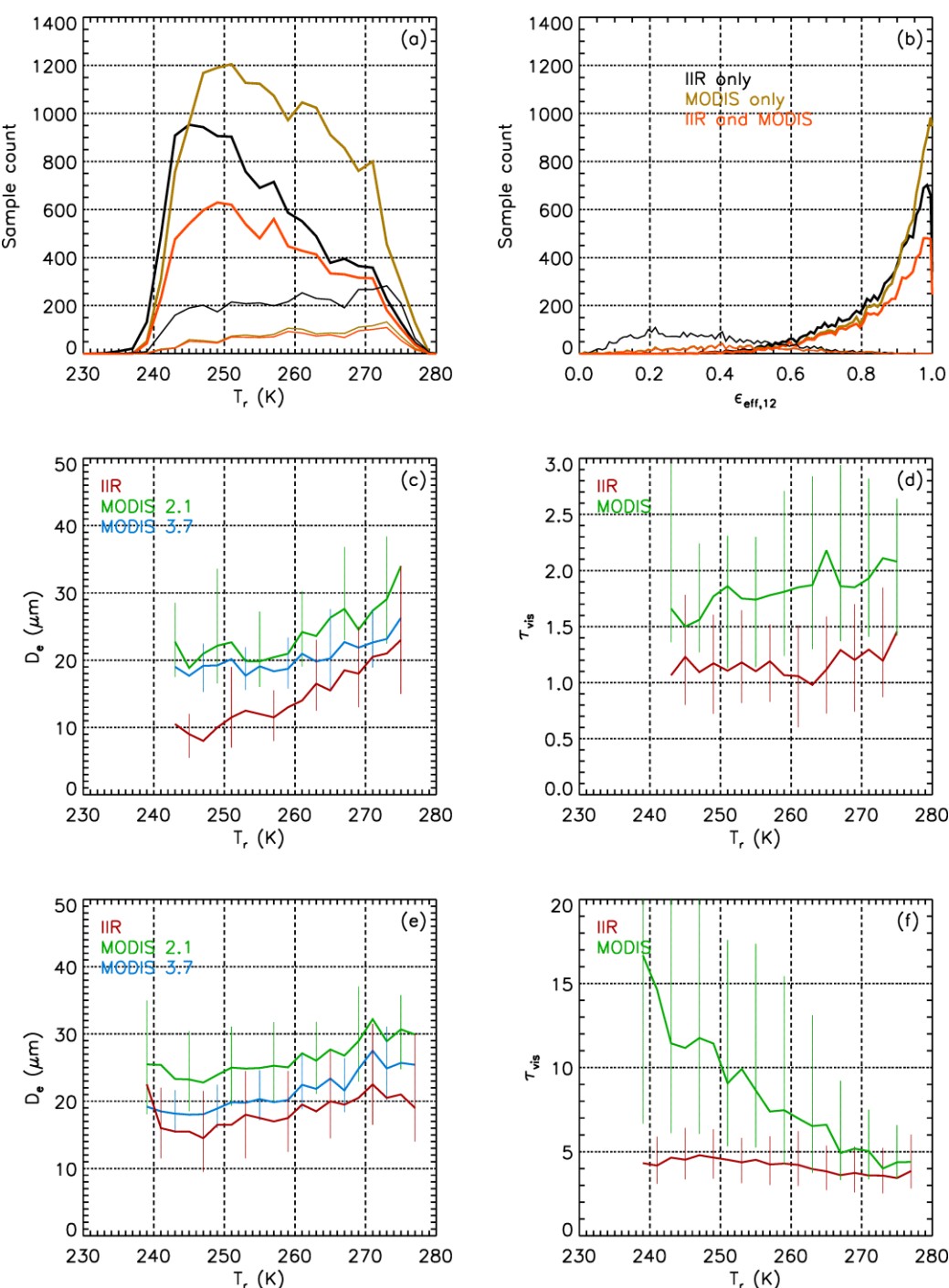

**Figure 17: IIR and MODIS comparisons over oceans between 60° S and 60° N in January 2008 for single-layered high confidence liquid**
**water clouds of centroid altitude > 4 km with MODIS water phase. Top: distributions of (a) IIR radiative temperature and (b) IIR**

**effective emissivity at 12.05 μm in ST (thin lines) and opaque (thick lines) clouds where IIR has confident retrievals (black), MODIS has successful retrievals at 2.1 and 3.7 μm (brown), and both IIR and MODIS retrievals are successful and can be compared (orange). Median $D_e$ vs. $T_r$ from IIR (red), MODIS 2.1 (green) and MODIS 3.7 (blue) in ST (c) and opaque (e) clouds; Median $\tau_{vis}$ from IIR (red) and MODIS (green) in ST (d) and opaque (f) clouds. The vertical bars in panels (c-f) are between the 25th and 75th percentiles.**

## 5 Conclusions and perspectives

This paper describes the impacts of the various changes implemented in the V4 IIR Level 2 algorithm on the effective emissivities and microphysical retrievals in ice clouds. We chose to illustrate and discuss the changes for one month's worth of data over ocean using a step-by-step approach so that data users can understand the differences and improvements that they should expect when using the recently released V4 IIR Level 2 data products. Retrievals in liquid water clouds, which were added in V4, are also presented. The IIR retrievals rely heavily on the scene classification reported for exactly co-located CALIOP observations. The results are presented for single-layer cases having the ocean surface as a reference and for which the CALIOP cloud classification and ice-water phase identification are determined with high confidence.

We show that in tenuous ST clouds, emissivity retrievals derived from both observed and computed background radiances are fully consistent in V4, whereas the inter-channel biases that were observed in V3 when the background radiance had to be computed introduced significant biases into the V3 microphysical retrievals. Our assessment is based on internal control criteria; i.e., the analysis of retrieved inter-channel effective emissivity differences at $\varepsilon_{eff,12} \sim 0$. Because the background radiance has to be computed for approximately 70 % of the retrievals in ST clouds, the number of unbiased emissivity retrievals is increased by a factor 3 in V4. In V4, the lowest effective emissivity for which microphysical retrievals are possible in more than 80 % of the pixels is reduced to ~0.05 (or $\tau_{vis} \sim 0.1$). In contrast, this lowest emissivity limit in V3 was as high as 0.25 in those cases of computed background radiances and was driven by the large biases in the 12/08 pair. Furthermore, when microphysical retrievals were possible in V3, the different 12-10 and 12-08 inter-channel biases induced large differences between the $D_e12/10$ and $D_e12/08$ diameters retrieved from the respective microphysical indices.

Perhaps one unique feature of the IIR algorithm is that the ice habit model is selected according to the relationship between the $\beta_{eff}12/10$ and $\beta_{eff}12/08$ inter-channel microphysical indices. In V4, the "TAMUice 2016" SCO (severely roughened single column) model is selected in 80 % of the cases in ST clouds at $T_r < 210$ K, and this fraction decreases at larger temperatures. The "TAMUice 2016" CO8 (severely roughened 8-element column aggregate) model is selected in 40 % of the cases when clouds have radiative temperatures larger than 230 K. In ice clouds, $D_e12/10$ is on average smaller than $D_e12/08$, with larger discrepancies below 230 K than above. Employing a technique similar to the IIR algorithm, Heidinger et al. (2015) also noticed differences between effective diameters retrieved from the Aqua/MODIS 32/31 and 31/29 pairs of channels when using the "TAMUice 2013" CO8 model (Yang et al., 2013), which was chosen for the MODIS Collection 6 data products for its consistency between visible and thermal infrared optical depth retrievals (Holz et al., 2016). We could not find a perfect agreement between $D_e12/10$ and $D_e12/08$ in liquid water clouds supposedly composed of spherical droplets. In the range of temperature between 240 K and 260 K, where both ice and liquid water clouds are found, $D_e12/10$ is larger than $D_e12/08$ in liquid water clouds while it is smaller in ice clouds, suggesting that these mismatches are not due to undetected residual biases in the IIR microphysical indices but instead to our LUTs. As noted earlier, the residual mismatch in liquid water clouds could be explained by inaccuracies in the refractive indices, which are taken constant whereas temperature-dependent indices have been reported (Zasetsky et al., 2005; Wagner et al., 2005). Likewise, the "TAMUice 2016" single-scattering properties are derived using refractive indices at 266 K (Warren and Brandt, 2008), but Iwabuchi and Yang (2011) reported that the temperature dependence of these properties in the thermal infrared is small but not negligible. While in V3 mismatches between IIR retrievals and the LUTs were largely due to inter-channel biases in the IIR retrievals, the improved accuracy in V4 opens the possibility for more detailed comparisons with the theory or modeling.

Retrievals in opaque ice clouds are improved in V4, especially at night and for 12/10 pair of channels, owing to corrections of the radiative temperature estimates. Refining the relationship between lidar geometric altitudes and infrared radiative temperature based on theoretical considerations (Part I) is deemed important per se, and quasi-perfectly co-located IIR and CALIOP observations offer a unique opportunity to test our theoretical approach. To make further progress in this topic and assess the V4 radiative temperature estimates in opaque clouds, the next step will be to use CloudSat extinction profiles from the lower parts of the clouds not seen by CALIOP.

Daytime comparisons with Aqua/MODIS Collection 6 data products are presented for co-located pixels where V4 IIR, MODIS 2.1 and MODIS 3.7 all have successful retrievals. This comparison demonstrated that IIR is best suited for retrievals in tenuous clouds of emissivity $< 0.2$ while MODIS is more efficient for denser clouds of emissivity $> 0.8$. IIR $D_e$ is in better agreement with MODIS 3.7 than with MODIS 2.1 in tropical ST ice clouds at $T_r < 200$ K. In contrast, IIR $D_e$ is in agreement with MODIS 2.1 in tropical opaque ice clouds at $T_r < 205$ K and in fair agreement with MODIS 3.7 at warmer temperatures. For opaque liquid water clouds having centroid altitudes greater than 4 km, so chosen to ensure sufficient radiative temperature contrast for the IIR retrievals, IIR $D_e$ is systematically smaller than MODIS 2.1 by 8 µm and smaller than MODIS 3.7 by 3 µm. The IIR technique appears to be perfectly suited for retrievals in ST supercooled liquid water clouds.

**Data availability**

The Version 3 IIR Level 2 track products used in this paper are available at https://doi.org/10.5067/IIR/CALIPSO/L2_Track-Beta-V3-01 (last access: 14 September 2020) and the Version 4 IIR Level 2 track products are available at https://doi.org/10.5067/CALIOP/CALIPSO/CAL_IIR_L2_Track-Standard-V4-20 (last access: 14 September 2020).

The IIR Level 2 track products are also available at the AERIS/ICARE Data and Services Center at https://www.icare.univ-lille.fr/data-access/data-archive-access/?dir=IIR/ (last access 14 September 2020).

Comparisons with MODIS Collection 6: co-location and MODIS visible/2.1 µm data are from the CALTRACK-5km_MYD06.v1.01 products which are available at the AERIS/ICARE Data and Services Center at https://www.icare.univ-lille.fr/data-access/data-archive-access/?dir=CALIOP/CALTRACK-5km_MYD06.v1.01/(last access: 14 September 2020). MODIS visible/3.7 µm data were extracted from the Collection 6 MYD06 products available at the AERIS/ICARE Data and Services Center at https://www.icare.univ-lille.fr/data-access/data-archive-access/?dir=MODIS/MYD06_L2.006/ (last access: 14 September 2020).

**Author contribution**

AG and JP defined the content and methodology of the paper and wrote the original draft. AG performed the data analysis and prepared the figures. NP was in charge of software development. MV provided assistance for the use of the CALIOP data. PD provided the FASRAD and FASDOM radiative transfer models and bulk scattering properties. PY provided the ice habit models from the "TAMUice 2016" database. DM provided the analytical functions derived from in situ measurements. All authors contributed to the review and editing of this paper.

## Competing interests

Author Jacques Pelon is a co-guest editor for the "CALIPSO Version 4 Algorithms and Data Products" special issue in Atmospheric Measurements Techniques but will not participate in any aspects of the editorial review of this manuscript. All other authors declare that they have no conflicts of interest.

## Acknowledgements

The authors are grateful to NASA LaRC, to SSAI (Science Systems and Applications, Inc.), to the Centre National d'Etudes Spatiales (CNES) and to Institut National des Sciences de l'Univers (INSU) for their support. We thank the AERIS infrastructure for providing access to the CALIPSO products, and for data processing during the development phase. We thank Brian Getzewich and Tim Murray for the processing of the Version 4 IIR Level 2 data at NASA LaRC.

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
