# Peer review of "Version 4 CALIPSO IIR ice and liquid water cloud microphysical properties, Part II: results over oceans"

_Atmospheric Measurement Techniques, 2020_

## Referee Comment (RC1) · Anonymous Referee #3 · 24 Jan 2021

Comments on AMT manuscript (amt-2020-388) entitled
"Version 4 CALIPSO IIR ice and liquid water cloud microphysical properties,
Part II: results over oceans" by Anne Garnier, Jacques Pelon, Nicolas Pascal, Mark
A. Vaughan, Philippe Dubuisson, Ping Yang, and David L. Mitchell.

This paper presents retrieval of cloud micro- and macro-physical properties over ocean using CALIOP Imaging Infrared Radiometer (IIR) Version 4 algorithm developed in Part I. It also shows the improvements over Version 3. However, there are several points to improve in the manuscript. The authors must revise their manuscript addressing my following specific comments.

Specific comments

1.  On p. 4, Fig. 1: 'Latitude (°C)' should be 'Latitude (°)'.

2.  On p. 5, lines 141-143: the authors state, "South of - 36.7° and down to - 37.2°, the portion of this cloud which is used as an opaque reference between - 36.45° and - 36.7° is included in a single opaque cloud of top altitude equal to 11.5 km, which extends down to the southernmost latitudes.". However, it seems to me that there is two-layer cloud between -36.7° and -37.2° in Fig. 1a. Why do the authors regard it as 'a single opaque cloud'?

3.  On p. 5, lines 155-157: the authors state, "In Fig. 1 we find cloud systems composed of ROI only (flag = 1), liquid water (WAT) only (flag = 2), ice and WAT (flag = 4), and some systems that include at least one layer of unknown phase (flag = 9)." 'flag=4' should be 'flag=6'.

4.  On p. 5, line 161: the authors state, "Effective emissivities in ST clouds vary between 0 and 0.9.". However, there are negative emissivities around -36.5° in Fig. 1f.

5.  On p. 6, lines 188-190: the authors state, "Scenes with only ST layers are spread into three main categories: only one layer, two vertically overlapping layers, and multi-layer configurations with two non-overlapping layers or more than two layers.". What do you mean by 'two non-overlapping layers'? Explain it briefly.

6.  On p. 8, lines 238-240: the authors state, "Overcorrections combined with uncertainties cause an increase of the fraction samples with $\varepsilon_{eff,12} > 1$, from 3 % in V3 to 12 % in V4 at night, and from 1.2 to 3.3 % for daytime data.". Does this sentence mean that V4 is worse than V3 in terms of overestimation of $\varepsilon_{eff,12}$?

7.  On p. 9, lines 265-267: the authors state, "This indicates residual inter-channel biases smaller than 0.1 K in V4 according to the simulations shown in Fig. 1c of Part I, which is consistent with the residual inter-channel differences seen in clear sky conditions (Part I)." 'Fig. 1c' should be 'Fig. 1b'.

8.  On p. 10, Fig. 4(a) and 4(b): 'color' is red (~4) at V3 $\tau_{vis}$ = V4 $\tau_{vis}$ = 1 in the whole plots, whereas 'color' is green (~2) at V3 $\tau_{vis}$ = V4 $\tau_{vis}$ = 1 in the embedded small plots. Why are these colors different at the same point? This comment is also applied to around V3 $\tau_{vis}$ = V4 $\tau_{vis}$ = 0.

9.  On p. 15, line 424: the authors state, "In this example, mean $D_e$ increases from 17 μm at 185 K to 53 μm at 245 K.". In comparison to Fig. 15 of Heymsfield and Iaquinta (2000), $D_e$ = 53 μm at 245 K is smaller than their observed ice crystal size around -35°C. How do the authors reconcile this difference?

10. On p. 18, Fig. 11(c) and 11(e): the same comment as the item #9 is applied to $D_e$ around $T_r$ = 245 K. The authors' retrieved $D_e$'s are smaller than MODIS 2.1 $D_e$. Does this mean that the authors' $D_e$'s are underestimated at higher temperatures?

11.  On p. 21, lines 574-576: the authors state, "Mean IIR De (Fig. 15b, red) increases steadily from 11 μm at 242 K to 18 μm at 270 K, while mean CALIOP particulate depolarization ratio (Fig. 15c) is constant and around 0.1.". However, Many researchers (e.g., Curry 1986, Garrett and Hobbs 1995, Nicholls and Leighton 1986, Noonkester 1984, Slingo et al. 1982, Stephens and Platt 1987) reported that cloud droplet effective radius increases from cloud base to cloud top. How the authors reconcile Fig. 15b with the opposite observations.

12. On p. 23, Fig. 16: The same comment as the item #11 is applied to Fig. 16(c). In other words, dependence of $D_e$ on temperature in Fig. 16(c) is opposite to the observed ones.

Technical corrections

1.  On p. 8, lines 235-236: the authors state, "these corrections have no to little impact for ST clouds". This sentence should be corrected.

2.  On p. 17, lines 484-485: the authors state, "This could explain than IIR $D_e$ is found…". 'than' should be 'that' in this sentence.

3.  On p. 27, line 759: the authors state, "in the 10-mm window region". '10-mm' should '10-μm' in this sentence.

4.  On p. 28, line 778: 'microphysics' should be 'Microphysics'.

5.  On p. 29, line 852: 'Minimis' should be 'Minnis'.

**References**

Curry, J. A., Interactions among turbulence, radiation, and microphysics in arctic stratus clouds, *J. Atmos. Sci.*, *43*, 90-106, 1986.

Garrett, T. J., and P. V. Hobbs, Long-range transport of continental aerosols over the Atlantic Ocean and their effects on cloud structure, *J. Atmos. Sci.*, *52*, 2977-2984, 1995.

Heymsfield, A. J., and J. Iaquinta, Cirrus Crystal Terminal Velocities, *J. Atmos. Sci.*, *57*, 916-938, 2000.

Nicholls, S., and J. Leighton, An observational study of the structure of stratiform cloud sheets, part 1, Structure, *Q. J. R. Meteorol. Soc.*, *112*, 431-460, 1986.

Noonkester, V. R., Droplet spectra observed in marine stratus cloud layers, *J. Atmos. Sci.*, *41*, 829-845, 1984.

Slingo, A., S. Nicholls, and J. Schmetz, Aircraft observation of marine stratocumulus during JASIN, *Q. J. R. Meteorol. Soc.*, *108*, 833-856, 1982.

Stephens, G. L., and C. M. R. Platt, Aircraft observations of the radiative and microphysical properties of stratocumulus and cumulus cloud fields, *J. Clim. Appl. Meteorol.*, *26*, 1243-1269, 1987.

---

## Referee Comment (RC2) · Anonymous Referee #4 · 29 Jan 2021

This is a review of the manuscript titled "Version 4 CALIPSO IIR ice and liquid water cloud microphysical properties, Part II: results over oceans" submitted to AMT by Garnier et al.

The manuscript is describes the updated version of the cloud retrieval products derived from the IIR instrument on CALIPSO. The manuscript is written well and contains some interesting results. However, as the paper aims to demonstrate the improved accuracy, some more comparisons with previously published results and some more discussion on the results is needed.

Some specific papers I suggest to reference are Kahn et al. (2018), King et al. (2013),

[Figure]

Platnick et al. (2017) and Van Diedenhoven et al. (2020). Detailed references are below. Further specific comments are below.

Section 2: Please add a legend at panels c and d of Figure 1.

Section 3. Could you remind the reader what the ice model used for V3 was and also that change caused by the ice model are discussed in part I?

Section 3.4.1: Kahn et al. (2018) also found a similar difference in effective radius of semi-transparant and opaque clouds with similar, but slightly larger, sizes. For opaque cloud tops, the global statistics of Van Diedenhoven et al. (2020) show similar, but somewhat larger, mean effective sizes over ocean. Furthermore, King et al. 2013 and Platnick et al. (2017) show similar histograms with seemingly comparable results.

Section 3.4.2: I find Figure 9 interesting and strongly suggest to also include a similar figure with the opaque cloud results in the paper.

Kahn et al. (2018) find similar variations of effective radius with cloud temperature for transparent clouds.

For thick clouds, Van Diedenhoven et al. (2020) also show similar variations of effective radius with cloud temperature and find evidence that these size variations are related to variations in crystal growth rates. They also show a vertical variation of crystal shape, which may be consistent with the SCO fraction shown in the manuscript. Please show the statistics also for thick clouds and discuss how it compares to Kahn et al. (2018) and Van Diedenhoven et al. (2020).

Since both habit and size vary with temperature, it may be interesting to investigate how habit varies with size. Again, such results can be compared to Van Diedenhoven et al. (2020) for thick clouds. Note that this is just a suggestion.

Section 3.5: Van Diedenhoven et al. (2020) found a vertical variation in ice asymmetry parameter that leads to a high bias in MODIS collection 6 ice effective radius for warm clouds. This may partly explain the larger differences between MODIS and IIR for the

warmest ice cloud tops.

Section 4.3.1: King et al. (2013) and Platnick et al. (2017) show similar histograms for liquid cloud tops. Please discuss the comparison.

Section 4.3.2: Although I'm not aware of any other published statistics of liquid drop effective radius as a function of cloud top height for (near-) global clouds, I find the decrease of effective radius with decreasing cloud top temperature a bit surprising. I would expect an increase of effective radius as, under an adiabatic assumption, drops grow as the clouds deepen. I do notice that MODIS results in Fig. 16 show the same variation. Please add some discussion about this in the text. Some more discussion on how these results relate to other results would be good.

References:

Kahn, B. H., Takahashi, H., Stephens, G. L., Yue, Q., Delanoë, J., Manipon, G., et al. (2018). Ice cloud microphysical trends observed by the atmospheric infrared sounder. Atmospheric Chemistry and Physics, 18(14), 10,715–10,739. https://doi.org/10.5194/acp‐18‐10715‐2018

Platnick, S., Meyer, K. G., King, M. D., Wind, G., Amarasinghe, N., Marchant, B., et al. (2017). The MODIS cloud optical and microphysical products: Collection 6 updates and examples from Terra and Aqua. IEEE Transactions on Geoscience and Remote Sensing, 55(1), 502–525. https://doi.org/10.1109/TGRS.2016.2610522

King, M.d., S. Platnick, W. P. Menzel, S. A. Ackerman, and P. A. Hubanks, "Spatial and temporal distribution of clouds observed by MODIS onboard the terra and aqua satellites," IEEE Trans. Geosci. Remote Sens., vol. 51, no. 7, pp. 3826–3852, 2013.

Van Diedenhoven, B., A.S. Ackerman, A.M. Fridlind, B. Cairns, and J. Riedi, 2020: Global statistics of cloud top ice microphysical and optical properties. J. Geophys. Res. Atmos., 125, no. 6, e2019JD031811, doi:10.1029/2019JD031811.

---

## Author Comment (AC1) · 26 Feb 2021

Response to anonymous referee #3

The authors are thankful to the referee for his/her thorough review of the paper.

All the comments have been taken into account, as detailed below, and the manuscript will be revised accordingly. In the following, the reviewer's comments are in black, and our answer to each comment is in red.

This paper presents retrieval of cloud micro- and macro-physical properties over ocean using CALIOP Imaging Infrared Radiometer (IIR) Version 4 algorithm developed in Part I. It also shows the improvements over Version 3. However, there are several points to improve in the manuscript. The authors must revise their manuscript addressing my following specific comments.

Specific comments

1   On p. 4, Fig. 1: 'Latitude (°C)' should be 'Latitude (°)'.
Fixed.

2   On p. 5, lines 141-143: the authors state, "South of - 36.7° and down to - 37.2°, the portion of this cloud which is used as an opaque reference between - 36.45° and - 36.7° is included in a single opaque cloud of top altitude equal to 11.5 km, which extends down to the southernmost latitudes.". However, it seems to me that there is two-layer cloud between -36.7° and -37.2° in Fig. 1a. Why do the authors regard it as 'a single opaque cloud'?
The authors confirm that, as written at the beginning of the section, the classification is provided by the V4 CALIOP cloud and aerosol 5-km layer products. In this particular case, the CALIOP algorithm identified only one layer. More details are available in section 3.2.5.3. (Closing Gaps Between Features) in the CALIOP layer detection ATBD, which is available at https://www-calipso.larc.nasa.gov/resources/pdfs/PC-SCI-202_Part2_rev1x01.pdf.

3   On p. 5, lines 155-157: the authors state, "In Fig. 1 we find cloud systems composed of ROI only (flag = 1), liquid water (WAT) only (flag = 2), ice and WAT (flag = 4), and some systems that include at least one layer of unknown phase (flag = 9)." 'flag=4' should be 'flag=6'.
Fixed.

4   On p. 5, line 161: the authors state, "Effective emissivities in ST clouds vary between 0 and 0.9.". However, there are negative emissivities around -36.5° in Fig. 1f.
Thank you for pointing this out. We added the following sentence:

*"The only exception is between -36.45° and -36.52°, where non-physical negative effective emissivities are retrieved because the computed background radiances are smaller than the observed radiances, and are therefore underestimated. In this case, the reference is an opaque cloud (see area highlighted in red in Fig. 1b), which is likely not sufficiently dense to behave as a blackbody source".*

5   On p. 6, lines 188-190: the authors state, "Scenes with only ST layers are spread into three main categories: only one layer, two vertically overlapping layers, and multi-layer configurations with two non-overlapping layers or more than two layers.". What do you mean by 'two non-overlapping layers'? Explain it briefly.

The text now reads (new text in italic):

"Scenes with only ST layers are spread into three main categories: only one layer, two vertically overlapping layers *detected at different horizontal averaging resolutions where the top altitude of the lower layer is greater than the base altitude of the higher layer*, and multi-layer configurations with two non-overlapping layers or more than two layers."

6     On p. 8, lines 238-240: the authors state, "Overcorrections combined with uncertainties cause an increase of the fraction samples with $\varepsilon_{eff,12} > 1$, from 3 % in V3 to 12 % in V4 at night, and from 1.2 to 3.3 % for daytime data.". Does this sentence mean that V4 is worse than V3 in terms of overestimation of $\varepsilon_{eff,12}$?

In opaque ice clouds, $\varepsilon_{eff,12}$ was underestimated in V3 and is increased in V4, with V4 distributions peaking at 0.99 and 0.97 for nighttime and daytime data, respectively, that is very close to 1. Because of unavoidable random uncertainties and possible overcorrections, the consequence is that the fraction of samples where $\varepsilon_{eff,12}$ is larger than 1 is increased in V4.

The text now reads (changes in italic):

"Nighttime and daytime $\varepsilon_{eff,12}$ distributions peak at larger $\varepsilon_{eff,12}$ in V4 ($\varepsilon_{eff,12} = 0.99$ and 0.97, respectively) than in V3 ($\varepsilon_{eff,12} = 0.94$). *Consequently, random uncertainties and possible overcorrections cause an increase of the fraction samples with $\varepsilon_{eff,12} > 1$, from 3 % in V3 to 12 % in V4 at night, and from 1.2 to 3.3 % for daytime data.*"

7. On p. 9, lines 265-267: the authors state, "This indicates residual inter-channel biases smaller than 0.1 K in V4 according to the simulations shown in Fig. 1c of Part I, which is consistent with the residual inter-channel differences seen in clear sky conditions (Part I)." 'Fig. 1c' should be 'Fig. 1b'.

Fixed.

8     On p. 10, Fig. 4(a) and 4(b): 'color' is red (~4) at V3τvis = V4τvis = 1 in the whole plots, whereas 'color' is green (~2) at V3τvis = V4τvis = 1 in the embedded small plots. Why are these colors different at the same point? This comment is also applied to around V3τvis = V4τvis = 0.

In the whole plots, the minimum τvis was -0.5 and not 0, which introduced confusion. The revised figure is:

[Figure]

The bin sizes are 0.2 for the whole plots, and 0.02 for the small plots. This is now clarified in the text where we added the following sentence:

*"The large plots where $\tau_{vis}$ ranges between 0 and 15 are built using bins equal to 0.2, and the embedded small plots show details for $\tau_{vis}$ smaller than 1 and bins equal to 0.02."*

9. On p. 15, line 424: the authors state, "In this example, mean De increases from 17 µm at 185 K to 53 µm at 245 K.". In comparison to Fig. 15 of Heymsfield and Iaquinta (2000), De = 53 µm at 245 K is smaller than their observed ice crystal size around -35°C. How do the authors reconcile this difference?

The $D_e$ and $T_r$ parameters characterize a cloud layer. In semi-transparent clouds of optical depth smaller than about 3, IIR $D_e$ is a layer "average" effective diameter. In opaque clouds, it is mostly representative of the portion of the cloud seen by CALIOP before the signal is totally attenuated. Thus, comparisons with in-cloud vertically resolved measurements are not straightforward. In Fig. 15 of Heymsfield and Iaquinta (2000), our understanding is that the ice crystals observed around -35 °C were near the base of the cloud layer, at about 7.5 km, while the top altitude was around 10.5 km. Because IIR retrievals characterize a layer or the upper part of a layer, depending on cloud optical depth, they might differ from observations near cloud bases.

10. On p. 18, Fig. 11(c) and 11(e): the same comment as the item #9 is applied to De around Tr = 245 K. The authors' retrieved De's are smaller than MODIS 2.1 De. Does this mean that the authors' De's are underestimated at higher temperatures?

This means that V4 IIR $D_e$ is smaller than MODIS 2.1 and 3.7. We added the following sentence:

*"At $T_r$ > 220 K, IIR $D_e$ is around 50-60 µm and smaller than both MODIS 2.1 and 3.7. We note that the agreement with MODIS would be improved using the parameterized functions derived from the unmodified in-situ PSDs that were presented in Sect. 3.4.3."*

11. On p. 21, lines 574-576: the authors state, "Mean IIR De (Fig. 15b, red) increases steadily from 11 µm at 242 K to 18 µm at 270 K, while mean CALIOP particulate depolarization ratio (Fig. 15c) is constant and around 0.1.". However, Many researchers (e.g., Curry 1986, Garrett and Hobbs 1995, Nicholls and Leighton 1986, Noonkester 1984, Slingo et al. 1982, Stephens and Platt 1987) reported that cloud droplet effective radius increases from cloud base to cloud top. How the authors reconcile Fig. 15b with the opposite observations.

In this study, cloud centroid altitude is deliberately chosen > 4 km. The water clouds are in the free troposphere and most of them are composed of supercooled droplets. For this figure, the clouds are semi-transparent and median optical depth is only 0.9. As for ice clouds, the $D_e$ and $T_r$ parameters characterize a cloud layer. We could not find references with similar statistics for a similar population of clouds. We commented on the increase of $D_e$ with temperature by adding the following sentence:

*"It seems that these thin clouds would not be associated with strong updrafts, and the increase of layer average $D_e$ with layer radiative temperature could indicate growth through vapor deposition. In addition, there is an increasing probability for supercooled droplets to freeze as temperature decreases and as their size increases."*

For more clarity regarding this population of water clouds, we added median effective emissivity and median optical depth in Table 5, and added the following sentence after the presentation of Fig. 15 (previous Fig. 14, now Fig. 15 after comments by referee #4):

*"Note that the IIR retrievals shown in Fig. 15 are for a population of very thin water clouds: median $\tau_{vis}$ is only 0.9 in ST clouds and between 4 and 5 in opaque clouds. "*

12. On p. 23, Fig. 16: The same comment as the item #11 is applied to Fig. 16(c). In other words, dependence of De on temperature in Fig. 16(c) is opposite to the observed ones.

See our answer to item #11.

Note that in Fig. 16c and 16e (now Fig. 17c and 17e), both MODIS and IIR see an increase of $D_e$ with temperature. We tried to improve the text, which now reads (changes in italic):

"As seen in Fig. 17a, $T_r$ spans between 235 K and 280 K, *and most of these sampled clouds are composed of supercooled droplets*. In ST clouds, *the three datasets show an increase of median $D_e$ (Fig. 17c) as $T_r$ increases from 243 K to 270 K, but with different slopes*: IIR $D_e$ increases with $T_r$ from 10 to 20 µm whereas both MODIS 2.1 and 3.7 are larger than about 20 µm. *As seen in Fig. 17d, these supercooled water clouds have optical depths between 1 and 2*, with MODIS $\tau_{vis}$ overestimating IIR $\tau_{vis}$ by about 50 %."

Technical corrections

1. On p. 8, lines 235-236: the authors state, "these corrections have no to little impact for ST clouds". This sentence should be corrected.

The new text reads: "*these corrections have essentially no impact for ST clouds*".

2. On p. 17, lines 484-485: the authors state, "This could explain than IIR De is found…". 'than' should be 'that' in this sentence.

Fixed.

3. On p. 27, line 759: the authors state, "in the 10-mm window region". '10-mm' should '10-µm' in this sentence.

Fixed.

4. On p. 28, line 778: 'microphysics' should be 'Microphysics'.

Fixed.

5. On p. 29, line 852: 'Minimis' should be 'Minnis'.

Fixed.

---

## Author Comment (AC2) · 26 Feb 2021

Response to anonymous referee #4

The authors are thankful to the referee for his/her thorough review of the paper.

Our responses are detailed below, and the manuscript has been revised accordingly. In the following, the reviewer's comments are in black, and our answer to each comment is in red.

The manuscript describes the updated version of the cloud retrieval products derived from the IIR instrument on CALIPSO. The manuscript is written well and contains some interesting results. However, as the paper aims to demonstrate the improved accuracy, some more comparisons with previously published results and some more discussion on the results is needed.
Some specific papers I suggest to reference are Kahn et al. (2018), King et al. (2013), Platnick et al. (2017) and Van Diedenhoven et al. (2020). Detailed references are below. Further specific comments are below.

Section 2: Please add a legend at panels c and d of Figure 1.
Done. The revised figure 1 is:

[Figure]

Section 3. Could you remind the reader what the ice model used for V3 was and also that change caused by the ice model are discussed in part I?
The following text has been added:

*"In V3, the LUTs were derived using single scattering properties of the "solid column" and "aggregate" crystal models from the database described in Yang et al. (2005), with no particle size distribution. We showed in Part I that everything else being equal, the size distribution introduced in V4 increases retrieved $D_e$."*

New reference:
Yang, P., Wei, H., Huang, H. L., Baum, B. A., Hu, Y. X., Kattawar, G. W., Mishchenko, M. I., and Fu, Q.: Scattering and absorption property database for non-spherical ice particles in the near-through far-infrared spectral region, Appl. Opt., 44, 5512–5523, https://doi.org/10.1364/AO.44.005512, 2005.

Section 3.4.1: Kahn et al. (2018) also found a similar difference in effective radius of semi-transparent and opaque clouds with similar, but slightly larger, sizes. For opaque cloud tops, the global statistics of Van Diedenhoven et al. (2020) show similar, but somewhat larger, mean effective sizes over ocean. Furthermore, King et al. 2013 and Platnick et al. (2017) show similar histograms with seemingly comparable results.
a)Note that the "non-opaque" clouds in Kahn et al. (2018) have emissivity < 0.98, and likely include our ST clouds and a very large fraction of our opaque clouds, because the term "opaque" in our study refers to clouds that are opaque to the CALIOP lidar. The CALIOP opaque clouds have emissivity typically > 0.8 and a lot of them have emissivity < 0.98 and are called "non-opaque" in Kahn et al. (2018). We added a reference to Guignard et al. (2012) who show variations of $D_e$ with effective emissivity.

The following text is added:

*"Median $D_e$ in opaque clouds is around 60 µm and the distributions peak at 50 µm. It is larger than in ST clouds, which is consistent with retrievals based on AIRS thermal infrared data (Guignard et al., 2012; Kahn et al., 2018)."*

New reference:

Guignard, A., Stubenrauch, C. J., Baran, A. J., and Armante, R.: Bulk microphysical properties of semi-transparent cirrus from AIRS: a six year global climatology and statistical analysis in synergy with geometrical profiling data from CloudSat-CALIPSO, Atmos. Chem. Phys., 12, 503–525, https://doi.org/10.5194/acp-12-503-2012, 2012.

b)We decided to reference Van Diedenhoven et al. (2020) in section 3.4.2 (see response to next comment), where Kahn et al. (2018) is referenced again.

c)We agree that comparisons with MODIS are important, which is why the paper includes a dedicated section (3.5) with comparisons of collocated MODIS and IIR retrievals for ST and opaque ice clouds. Platnick et al. (2017) is referenced in this section. We chose not to reference King et al. (2013) because we wanted to focus on comparisons with MODIS Collection 6.

As discussed in Section 3.5, IIR and MODIS retrievals typically do not cover the same range of optical depths. To clarify the range of optical depths covered by IIR, we added median $\varepsilon_{eff,12}$ and median $\tau_{vis}$ in Table 4. Furthermore, we added the following sentences in section 3.4.1:

*"The ST clouds are optically thin, with median IIR $\tau_{vis}$ of only 0.2 - 0.26."*

and

*"These opaque clouds have median $\varepsilon_{eff,12}$ equal to 0.95 at night but only 0.86 for daytime data, with median IIR $\tau_{vis}$ equal to 5.6 and 3.8, respectively"*.

Section 3.4.2: I find Figure 9 interesting and strongly suggest to also include a similar figure with the opaque cloud results in the paper.

We followed the reviewer's suggestion and included a new figure (Fig. 10) for opaque clouds (all the following figures are renumbered). The new Figure 10 is:

[Figure]

**Fig. 10: Same as Fig. 9 but for opaque clouds.**

Section 3.4.2 now reads as follows. The changes are in italic.

"Recall that $D_e$ is retrieved using the crystal model (SCO or CO8) that agrees the best with IIR in terms of the relationship between $\beta_{eff}12/10$ and $\beta_{eff}$ 12/08. As seen in Fig. 9, the SCO crystal model is selected in 80 % of the ST clouds of $T_r < 205$ K. This fraction steadily decreases down to 60 % as $T_r$ increases up to 230 K (Fig. 9b) and remains stable above 230 K. This result is qualitatively consistent with previous findings using V3 (Garnier et al., 2015), and, as was discussed in this paper, both the IIR model selection and the mean CALIOP integrated particulate depolarization ratio (in black in Fig. 9b) indicate changes of crystal habit with temperature. *In opaque clouds (Fig. 10), both the IIR model selection and the CALIOP depolarization ratio between 200 K and 230 K are less temperature-dependent than in ST clouds*. The difference between mean $D_e12/10$ and mean $D_e12/08$ in black and grey in Fig. 9c is a measure of the residual mismatch between IIR observations and the selected model. We see two temperature regimes, that is, below and above 225 K, with a better agreement between IIR and the LUTs at the warmer temperatures. This suggests that the V4 models are better suited for warmer clouds and that they do not perfectly reproduce the infrared spectral signatures of colder clouds composed of small crystals. It is acknowledged that the highly variable ice particle shapes found in ice clouds (Lawson et al., 2019 and references therein) are likely not fully reproduced through the two models chosen for the V4 algorithm. It is further noted that the Clouds and the Earth's Radiant Energy System (CERES) science team is planning to use a two-habit model for

retrievals in the visible/near infrared spectral domain (Liu et al., 2014; Loeb et al., 2018). This model would be a mixture of two habits (single column and an ensemble of aggregates) whose mixing ratio would vary with ice crystal maximum dimension, with single columns prevailing for the smaller dimensions. Interestingly, our findings appear to be consistent with this approach.

*In both thin ST clouds (Fig. 9c) and opaque clouds (Fig. 10c), $D_e$ increases with cloud radiative temperature until it reaches a maximum value around 250 K in ST clouds and 230 K in opaque clouds. Kahn et al. (2018) found that for clouds of emissivity smaller than 0.98, $D_e$ is maximum and around 50 µm at 230 K, which is consistent with our findings, keeping in mind that clouds with emissivity smaller than 0.98 are found in both our ST and opaque clouds. The increase of cloud average $D_e$ with cloud radiative temperature in ST clouds (Fig. 9c) is in general agreement with numerous previous findings (e.g. Hong and Liu, 2015).* The decrease *of $D_e$* between $T_r$ = 250 K and 260 K *for ST clouds* is possibly due to an increasing fraction of small liquid droplets in these prevailingly ice layers, which would be consistent with the fact that CALIOP integrated particulate depolarization ratio decreases from 0.37 to 0.30 (Fig. 9b). *Similar comments apply for opaque clouds for $T_r$ between 230 and 260 K. Using combined POLDER (POLarization and Directionality of the Earth's Reflectances) and MODIS data, Van Diedenhoven et al. (2020) found that $D_e$ at the top of thick clouds of optical depth larger than 5 is maximum at cloud top temperature equal to 250 K, rather than $T_r$ = 230 K for our opaque clouds. This discrepancy might be partly explained if the cloud radiative altitude is higher in the cloud than the cloud top derived from the visible observations, which could also explain that $D_e$ shown in Van Diedenhoven et al. (2020) is larger than in this study."*

Kahn et al. (2018) find similar variations of effective radius with cloud temperature for transparent clouds. Our understanding is that they find similar variations for non-opaque clouds, that they define as clouds of emissivity < 0.98, which corresponds to our ST clouds and a large fraction of our opaque clouds.

For thick clouds, Van Diedenhoven et al. (2020) also show similar variations of effective radius with cloud temperature and find evidence that these size variations are related to variations in crystal growth rates. They also show a vertical variation of crystal shape, which may be consistent with the SCO fraction shown in the manuscript. Please show the statistics also for thick clouds and discuss how it compares to Kahn et al. (2018) and Van Diedenhoven et al. (2020).
We added a new Fig. 10 with results for opaque clouds as suggested and included a discussion (see above).

Since both habit and size vary with temperature, it may be interesting to investigate how habit varies with size. Again, such results can be compared to Van Diedenhoven et al. (2020) for thick clouds. Note that this is just a suggestion.
We are thankful to the reviewer for this great suggestion. We could carry out this type of analyses for a larger dataset for a future publication.

Section 3.5: Van Diedenhoven et al. (2020) found a vertical variation in ice asymmetry parameter that leads to a high bias in MODIS collection 6 ice effective radius for warm clouds. This may partly explain the larger differences between MODIS and IIR for the warmest ice cloud tops.
Again, we are thankful to the reviewer for this possible explanation.

Section 4.3.1: King et al. (2013) and Platnick et al. (2017) show similar histograms for liquid cloud tops. Please discuss the comparison.
We explain at the beginning of Section 4 that the water clouds selected for this study have centroid altitude larger than 4 km and that most of them are composed of supercooled droplets.

We added median $\varepsilon_{eff,12}$ and median $\tau_{vis}$ in Table 5 to highlight the range of optical depths covered by IIR retrievals. Furthermore, the following sentences are now added (the new Fig. 15 is the previous Fig. 14):

"*Note that the IIR retrievals shown in Fig. 15 are for a population of very thin water clouds: median $\tau_{vis}$ is only 0.9 in ST clouds and between 4 and 5 in opaque clouds*"

IIR histograms shown in Fig. 15 and MODIS histograms found in the literature for liquid cloud tops are for different populations of clouds. The IIR cloud population is composed of thin supercooled water clouds in the free troposphere, while the MODIS cloud population includes warmer and lower clouds of larger optical depth. Therefore, we chose to have a dedicated section where we show comparisons of collocated IIR and MODIS retrievals (Section 4.4). We chose not to reference King et al. (2013) because we wanted to focus on comparisons with MODIS Collection 6.

We added this sentence at the beginning of Sect. 4 to inform the reader that comparisons with MODIS are shown in a dedicated section:
"*Our results are presented in Sect. 4.1 to 4.3 and comparisons with MODIS are shown in Sect. 4.4.*"

Section 4.3.2: Although I'm not aware of any other published statistics of liquid drop effective radius as a function of cloud top height for (near-) global clouds, I find the decrease of effective radius with decreasing cloud top temperature a bit surprising. I would expect an increase of effective radius as, under an adiabatic assumption, drops grow as the clouds deepen. I do notice that MODIS results in Fig. 16 show the same variation. Please add some discussion about this in the text. Some more discussion
on how these results relate to other results would be good.
As mentioned earlier, the water clouds selected for this study have centroid altitude larger than 4 km, most of them are composed of supercooled droplets, and in section 4.3.1, we now give the range of optical depths for ST and opaque clouds in Table 5.
We could not find references with similar statistics for a similar population of clouds. We commented on the increase of $D_e$ with temperature by adding the following:

"*It seems that these thin clouds would not be associated with strong updrafts, and the increase of layer average $D_e$ with layer radiative temperature could indicate growth through vapor deposition. In addition, there is an increasing probability for supercooled droplets to freeze as temperature decreases and as their size increases.*"